# Subquadratic Algorithms and Hardness for Attention with Any Temperature

**Shreya Gupta**
UW
Seattle, WA
sgupta63@cs.washington.edu

**Boyang Huang**
Yale
New Haven, CT
boyang.huang@yale.edu

**Barna Saha**
UCSD
La Jolla, CA
bsaha@ucsd.edu

**Yinzhan Xu**
UCSD
La Jolla, CA
xyzhan@ucsd.edu

**Christopher Ye**
UCSD
La Jolla, CA
czye@ucsd.edu

## Abstract

Despite the popularity of the Transformer architecture, the standard algorithm for computing Attention suffers from quadratic time complexity in context length $n$. Alman and Song showed that when the head dimension $d = \Theta(\log n)$, sub-quadratic Attention is possible if and only if the inputs have small entries bounded by $B = o(\sqrt{\log n})$ in absolute values, under the Strong Exponential Time Hypothesis (SETH). Equivalently, subquadratic Attention is possible if and only if the softmax is applied with high temperature for $d = \Theta(\log n)$. Running times of these algorithms depend exponentially on $B$ and thus they do not lead to even a polynomial-time algorithm outside the specific range of $B$.

This naturally leads to the question: when can Attention be computed efficiently without strong assumptions on temperature? Are there fast attention algorithms that scale polylogarithmically with entry size $B$? In this work, we resolve this question and characterize when fast Attention for arbitrary temperatures is possible. First, for all constant $d = O(1)$, we give the first subquadratic $\tilde{O}(n^{2-1/d} \cdot \mathrm{polylog}(B))$ time algorithm for Attention with large $B$. Our result holds even for matrices with large head dimension if they have low rank. Combined with a reduction from Gradient Computation to Attention, we obtain a subquadratic algorithm for the full LLM training process. Furthermore, we show that any substantial improvement on our algorithm is unlikely. In particular, we show that even when $d = 2^{\Theta(\log^* n)}$, Attention requires $n^{2-o(1)}$ time under SETH.

Finally, in the regime where $d = \mathrm{poly}(n)$, the standard algorithm requires $O(n^2 d)$ time while previous lower bounds only ruled out algorithms with truly subquadratic time in $n$. We close this gap and show that the standard algorithm is optimal under popular fine-grained complexity assumptions.

## 1 Introduction

Large Language Models powered by the Transformer architecture (Vaswani et al., 2017) have been at the heart of modern AI revolution completely reshaping the landscapes of natural language processing, computer vision, and multitude of other applications. The Attention mechanism is the cornerstone of the Transformer architecture. Attention computes correlations between different tokens of the sequences, allowing Transformers to model dependencies regardless of the position of the tokens in the sequences. Despite its popularity, standard algorithms for computing Attention require quadratic time complexity, as they compute the Attention matrix explicitly.

Formally, the Attention mechanism is defined as follows. Let $Q, K, V$ be size $n \times d$ matrices (respectively query, key and value matrices). We call $n$ the context length and $d$ the head dimension.

The Attention matrix is obtained by applying softmax[1] to each row of $QK^\top$. Each entry in the matrix represents the attention weight between a particular input token (query token $Q$) and output token (key token $K$). Finally, Attention outputs the product of the Attention matrix with $V$.

We give the formal definition below. Note that $\exp(X)$ applies $\exp$ to each entry of a matrix $X$.

**Definition 1.1** (Attention). Given input matrices $Q, K, V \in \mathbb{R}^{n \times d}$, *Attention* on $Q, K, V$ is defined $\mathrm{Attention}(Q, K, V) := D^{-1}AV \in \mathbb{R}^{n \times d}$ where $A := \exp(QK^\top)$[2] and $D := \mathrm{diag}(A\mathbf{1})$.

In practice, there is an input $X \in \mathbb{R}^{n \times d}$ and weight matrices $W_Q, W_K, W_V \in \mathbb{R}^{d \times d}$ such that $Q = XW_Q, K = XW_K, V = XW_V$. Since $Q, K, V$ can be computed from $X, W_Q, W_K, W_V$ in $O(nd^2)$ time, we assume for simplicity that the inputs $Q, K, V$ are given directly.

Typically, it suffices to *approximately* perform Attention computations. In particular, it is not necessary (or even reasonable) to expect Attention to be computed exactly due to the softmax operation. Thus, we study Approximate Attention, where each entry is computed with polynomial precision (i.e. inverse polynomial additive error).

**Definition 1.2** (Approximate Attention Computation $\mathsf{AttC}(n, d, B, \varepsilon)$). Given matrices $Q, K, V \in [-B, B]^{n \times d}$ and $B, \varepsilon > 0$, compute $O \in \mathbb{R}^{n \times d}$ such that $\|O - \mathrm{Attention}(Q, K, V)\|_\infty < \varepsilon$.

The standard (and most widely used) algorithm for Attention (even in approximate form) requires quadratic time. The algorithm begins by explicitly computing matrix product $QK^\top$, applies softmax to obtain $D^{-1}A$ and then computes the matrix product $(D^{-1}A)V$. Using standard matrix multiplication, this requires $O(n^2d)$ time. Even ignoring computation time of matrix multiplication, explicitly computing the $A$ matrix already requires $\Omega(n^2)$ time.

However, the inputs (and outputs) only have size $O(nd)$. Indeed, an algorithm that does not compute $A$ explicitly could compute Attention in $O(nd)$ time, incurring only *linear* dependence on the context length $n$. This leads to the fundamental question concerning the complexity of Attention.

*Question 1: When can Attention be computed faster than $n^2d$ time?*

Towards answering this question, Alman & Song (2024a) showed that for $d = \Theta(\log n)$, Attention can be computed in $n^{1+o(1)}$ time whenever $B = o(\sqrt{\log n})$. Furthermore, whenever $B = \Omega(\sqrt{\log n})$ and $d = \Theta(\log n)$, Attention requires $n^{2-o(1)}$ time under $\mathsf{SETH}$, a popular hardness hypothesis.

Yet there remain several shortcomings in our current understanding of Attention. Fast algorithms for Attention are only known for inputs with small entries (i.e. $B = o(\sqrt{\log n})$). Such a strong bound on the entries of $Q, K$ essentially restricts the Attention mechanism to use softmax with high temperature (enforcing a near-uniform distribution over the value matrix). Temperature, denoted by $T$, is a key hyperparameter for Attention that dictates how random the output is. Formally, Attention with temperature $T$ replaces $A := \exp(QK^\top)$ with $A := \exp(QK^\top/T)$ so that high temperature corresponds to high entropy (more likely to select keys with lower scores). In many tasks across machine learning, temperature is a key hyperparameter with potentially significant impact on accuracy and stability (Agarwala et al., 2023; Xuan et al., 2025). While the original transformer of Vaswani et al. (2017) fixes attention temperature to $\Theta(\sqrt{d})$ and only varies temperature in the final softmax layer, recent works have shown that varying attention temperature (in some cases even setting it as a learnable parameter) can improve performance across NLP and vision applications (Lin et al., 2018; Zhang et al., 2022; Peng et al., 2023; Lee et al., 2021; Chen et al., 2023; Dufter et al., 2020; Zou et al., 2024; Demir & Dogan, 2025). Indeed, Alman & Song (2025b) prove that transformers with high temperature are provably less expressive. Beyond transformers, in contrastive learning, temperature has been found to significantly impact both the accuracy (Chen et al., 2020; Wang & Liu, 2021; Hu et al., 2021) as well as the learned representations (Wang & Isola, 2020; Wang & Liu, 2021; Robinson et al., 2021) of the model. Dynamically varying temperature throughout the training process can also help balance multiple training objectives (Khaertdinov et al., 2022; Kukleva et al., 2023; Manna et al., 2023). In instances where low entropy is required, no subquadratic algorithms are known.

---

[1] Given a vector $x$, applying softmax to $x$ replaces $x_i$ with $\exp(x_i)/\sum_j \exp(x_j)$.

[2] In practice, a scaled dot-product attention, defined as $A := (QK^\top/\sqrt{d})$, is also commonly used for training efficiency Vaswani et al. (2017).

Furthermore, it is generally undesirable for the running time of an algorithm to scale poorly with the numerical values of the input. In fact for many fundamental problems (Knapsack, All-Pairs Shortest Paths, 3-SUM), having small entries makes the problems much easier. For example, there is a simple pseudo-polynomial time dynamic programming algorithm for Knapsack, while designing a polynomial time algorithm for Knapsack is NP-complete.[3] Therefore, in this work we study algorithms for Attention that scale polynomially with the *representation length* of the entries. Equivalently, the algorithm should scale polylogarithmically with the entry size $B$.

Currently, the only known algorithms for Attention beyond the standard $O(n^2 d)$ algorithm scales *exponentially* with the entry size $B$ (Alman & Song, 2024a). Following the terminology of pseudo-polynomial time, we will call an algorithm that is subquadratic but scaling polynomially (or worse) with the numerical value of the input pseudo-subquadratic. We call an algorithm that is subquadratic and scales logarithmically with the numerical value of the inputs (non-pseudo-)subquadratic, or simply subquadratic. Following from our above discussion, the question of whether subquadratic algorithms for Attention exist remains open.[4] Even if $d = O(1)$, there is a tantalizing gap between the $O(n^2)$ upper bound and the $\Omega(n)$ lower bound.

*Question 2: Is there a truly (non-pseudo-)subquadratic algorithm for Attention?[5]*

In our work, we resolve this question for almost all regimes of head dimension $d$. Our main result gives the first truly sub-quadratic algorithm for attention that scales polylogarithmically with entry-size $B$. Our algorithm obtains truly sub-quadratic time for constant $d$.[6]

**Theorem 1.1** (Main Theorem). *Let $d = O(1)$. There is an algorithm that computes $\mathsf{AttC}(n, d, B, \varepsilon)$ in $\tilde{O}(n^{2-1/d} \cdot \mathrm{polylog}(B/\varepsilon))$ time.*

The result also generalizes to the case where the matrices $Q, K$ have low rank.

**Theorem 1.2.** *Let $r = O(1)$. There is an $\tilde{O}\left(nd + n^{2-1/r} \cdot \mathrm{polylog}(B/\varepsilon)\right)$ time algorithm computing $\mathsf{AttC}(n, d, B, \varepsilon)$ where $r = \min(\mathrm{rank}(Q), \mathrm{rank}(K))$.*

As a side result, we complement this algorithm with a subquadratic algorithm for Attention Gradient Computation. In the training process, gradient descent tunes the weight matrices $W_Q, W_K, W_V$ according to the input data. In contrast to previous algorithms which give ad hoc algorithms for gradient computation, we show that gradient computation can be generically reduced to attention computation. Combined with our previous result, we give a truly (non-pseudo-)subquadratic algorithm for the full LLM training process when $d = O(1)$.

**Theorem 1.3** (Informal Theorem B.1). *The Attention gradient can be computed with $O(d)$ calls to $\mathsf{AttC}(n, d, B, \varepsilon/\Theta(ndB^3))$ with $O(nd^2)$ overhead. In particular, if $d = O(1)$ the Attention gradient can be computed in $\tilde{O}(n^{2-1/d}\mathrm{polylog}(B/\varepsilon))$ time.*

Above, we obtain a sub-quadratic algorithm for constant $d$. When $d = \omega(1)$ is super-constant, the above algorithms requires $n^{2-o(1)}$ time. Is there a truly subquadratic algorithm for super-constant $d$? Our remaining results provide stronger lower bounds for super-constant $d$. Alman & Song (2024a) show that $n^{2-o(1)}$ time is necessary when $d = \Omega(\log n)$ under the Strong Exponential Time Hypothesis (SETH). Under the same hardness assumption we provide a much stronger lower bound and show that Attention is hard even when $d = 2^{\Omega(\log^* n)}$.[7]

**Theorem 1.4** (Informal Theorem C.4). *Under SETH, $\mathsf{AttC}(n, d, B, \varepsilon)$ requires $n^{2-o(1)}$ time for $d = 2^{\Omega(\log^* n)}$ and $B = \mathrm{poly}(n)$.*

It suffices to consider instances with polynomial entry size $B = \mathrm{poly}(n)$ since any (non-pseudo-) subquadratic algorithm must handle such instances in subquadratic time. Formally, we show that

---

[3]An algorithm runs in pseudo-polynomial time if its running time is polynomial in the numerical value of the input. A polynomial time algorithm must be polynomial in the length of the input.

[4]Similarly, while there are pseudo-subcubic algorithms for APSP (e.g., Shoshan & Zwick (1999); Zwick (2002)), there is no truly subcubic ($O(n^{3-c})$ for some $c > 0$) algorithm.

[5]An algorithm runs in truly subquadratic time if it runs in $O(n^{2-c})$ time for some $c > 0$

[6]We use $\tilde{O}(\cdot)$ notation to suppress polylogarithmic factors.

[7]$\log^*$ denotes the iterated logarithm. For example, $\log^*(16) = 3$ since $\log \log \log 16 \leq 1$.

any fast algorithm for $\mathsf{AttC}(n, d, B, \varepsilon)$ implies a fast algorithm for (Bichromatic) Maximum Inner Product (Max-IP) on $d$-dimensional vectors with integer entries. The (Bichromatic) Max-IP problem asks an algorithm given two sets of vectors $A, B \subseteq \mathbb{Z}^d$ to compute $\max_{a \in A, b \in B} a \cdot b$. Under $\mathsf{SETH}$, this requires $n^{2-o(1)}$ time whenever $d = 2^{\Omega(\log^* n)}$ (Chen, 2018). Furthermore, the best known algorithms for Max-IP run in $n^{2-\Theta(1/d)}$ time (Yao, 1982; Agarwal et al., 1991; Matoušek, 1992) so that any algorithm improving significantly over Theorem 1.1 must improve upon the best known algorithms for Max-IP. Chen (2018) conjectures that no such algorithm exists under $\mathsf{SETH}$.

**Stronger Lower Bounds for Large Head Dimension.** The head dimension $d$ can often be relatively large with respect to the context length $n$ (in some cases e.g. Vaswani et al. (2017), the head dimension $d$ can even be larger than the context length $n$). In these settings, a large gap remains between the standard algorithm requiring $O(n^2 d)$ time and the known $n^{2-o(1)}$ lower bound. We address this gap and shows that the standard algorithm is conditionally optimal.

Our conditional lower bound depends on a natural generalization of a popular hypothesis. The Orthogonal Vectors (OV) problem is among the most well studied problems in fine-grained complexity. In the OV problem, an algorithm is given two sets of $n$ vectors $A, B \subseteq \{0, 1\}^d$ and is asked to determine if there exists an orthogonal pair $a \in A, b \in B$ such that $a \cdot b = 0$. The naive algorithm for this problem requires $O(n^2 d)$ time and the current best algorithm for OV achieves truly subquadratic time only for $d = O(\log n)$ (Abboud et al., 2015b; Chan & Williams, 2016). A central hypothesis (known as the OV Hypothesis) in fine-grained complexity states that there is no $n^{2-o(1)}$ algorithm for OV whenever $d = \omega(\log n)$, and the OV Hypothesis is known to hold under $\mathsf{SETH}$ (Williams, 2004).

If $d = \text{poly}(n)$, one can compute $a \cdot b$ for all pairs $a \in A, b \in B$ using a matrix product between an $n \times d$ matrix containing the vectors in $A$ as rows and a $d \times n$ matrix containing the vectors of $B$ as columns. The above algorithm requires $O(\mathsf{T}_{\mathsf{MUL}}(n, d, n))$ time, where $\mathsf{T}_{\mathsf{MUL}}(a, b, c)$ is the time complexity for multiplying an $a \times b$ matrix with a $b \times c$ matrix. The High-Dimensional OV Hypothesis introduced by Dalirrooyfard & Kaufmann (2021) hypothesized that when $d = n$, any algorithm computing OV requires $\mathsf{T}_{\mathsf{MUL}}(n, n, n)^{1-o(1)} = n^{\omega-o(1)}$ time, where $\omega < 2.3714$ denotes the square matrix multiplication exponent (Alman et al., 2025). We consider a generalization of their hypothesis: the $\mathsf{T}_{\mathsf{MUL}}(n, d, n)^{1-o(1)}$ running time is required for any $d = \text{poly}(n)$. We call it the Generalized High-Dimensional OV Hypothesis.

Under this hypothesis, we show that the standard algorithm for computing Attention is optimal. Note that using fast matrix multiplication, one can easily obtain an algorithm for Attention using $O(\mathsf{T}_{\mathsf{MUL}}(n, d, n))$ time.

**Theorem 1.5** (Informal Theorem C.5). *Under the Generalized High-Dimensional OV Hypothesis, $\mathsf{AttC}(n, d, B, \varepsilon)$ requires $\mathsf{T}_{\mathsf{MUL}}(n, d, n)^{1-o(1)}$ time for $d = \text{poly}(n)$.*

Table 1 summarizes our results. In particular, we tightly characterize the complexity of Attention (up to sub-polynomial factors) when $B = \text{poly}(n)$ for all regimes of $d$ except $1 \ll d \ll 2^{\Theta(\log^* n)}$. Within this regime, our running time matches the best known algorithms for Max-IP (Yao, 1982; Agarwal et al., 1991; Matoušek, 1992), and as mentioned earlier, significant improvements over our algorithm will imply improvements over the current best known algorithms for Max-IP which will be a breakthrough.

## 1.1 TECHNICAL OVERVIEW

In this section, we give a high level overview of our algorithm. For simplicity, we focus on the $d = 1$ case in this overview. Given inputs $q, k, v \in \mathbb{R}^n$, our goal is to compute $o_i = \sum_j p_{i,j} v_j$ for all $i$ where $p_{i,j}$ are probabilities in the softmax distribution proportional to $\exp(q_i k_j)$.

Our first observation is that small key values can be discarded: in particular, we show that for each $i$ it suffices to only consider keys where $q_i k_j$ is near the maximum. Assume without loss of generality that $q_i > 0$ and let $k_{\max} = \max_j k_j$. For an appropriate threshold $t$, we define $j$ to be *irrelevant* (with respect to $q_i$) if $q_i k_j \leq q_i k_{\max} - t$ and *relevant* otherwise. By setting $t = \Theta(\log(n/\varepsilon))$, we can ensure that all softmax probabilities corresponding to irrelevant indices are negligible. Since discarding such $j$ does not significantly change the value of the output significantly, we consider only relevant $j$ for the remainder of the overview.

Table 1: Summary of known results when $B = \text{poly}(n)$ and $\varepsilon = 1/\text{poly}(n)$. Sub-polynomial dependencies are suppressed for simplicity. Previous upper bounds that are not starred follow from the standard algorithm for computing attention (Vaswani et al., 2017). Previous lower bounds that are not starred are trivial and follow directly from input and output size. $*$ The starred results are due to Alman & Song (2024a). For $d = \Theta(\log n)$, their lower bound holds when $B = \Omega(\sqrt{\log n})$ while ours holds even when $B \geq \log 2$.

| | Upper Bound | | Lower Bound | |
|---|---|---|---|---|
| $d$ | Previous | New | Previous | New |
| $O(1)$ | $n^2$ | $n^{2-1/d}$ (1.1) | $n$ | |
| $2^{\Theta(\log^* n)}$ | $n^2$ | | $n$ | $n^{2-o(1)}$ (1.4) |
| $\Theta(\log n)$ | $n^2$ | | $n^{2-o(1)}*$ | $n^{2-o(1)}$ (C.7) |
| $\text{poly}(n)$ | $\mathsf{T}_{\mathsf{MUL}}(n, d, n)$ | | $n^{2-o(1)}*$ | $\mathsf{T}_{\mathsf{MUL}}(n, d, n)^{1-o(1)}$ (1.5) |

Figure 1: Rounding based algorithm for 1-dimensional Attention illustrated for $q_i = 1$. Each point is placed at $k_j$ and has value $v_j$. Points (e.g. $k_1$) such that $q_i k_j < q_i k_{\max} - t$ are irrelevant and discarded (in this example $q_i k_{\max} - t = 1$). Relevant points with similar $k_j$ (e.g. $\{k_2, k_3\}$ or $\{k_6, k_7, k_8\}$) are grouped together and assigned the same (rounded) key $\overline{k}$. The width of each region is $\log(1 + \varepsilon)$ (in this example $\log(1 + \varepsilon) = 3$). The algorithm outputs $\sum \overline{p}_j v_j$ where $\overline{p}_j \propto \exp(\overline{k}_j)$.

Combining this observation with a simple rounding scheme, we already obtain a modest improvement over known algorithms for Attention. We illustrate this for the $d = 1$ case. Consider a relevant key $k_j$. If we round such $k_j$ to $\overline{k}_j$ such that $q_i k_j \leq q_i \overline{k}_j \leq q_i k_j + \log(1 + \varepsilon)$, then $e^{q_i \overline{k}_j}$ is a $(1 + \varepsilon)$-multiplicative approximation of $e^{q_i k_j}$. This gives us good multiplicative approximations of the softmax probabilities. Plugging in these approximate probabilities, we obtain a good multiplicative approximation of the output.

Since the value of the output is bounded by entries of the value matrix $V$, (i.e. $o_i = O(B)$), this gives a $\varepsilon B$-additive approximation of the output. To compute the approximation, we can now treat all keys $k_j$ that are rounded to the same value $\overline{k}_j$ as equivalent. Since relevant keys are within a range of length $t$ and we round all keys within $\log(1 + \varepsilon)$ to the same value, we only need to consider $O(t/\log(1 + \varepsilon)) = \tilde{O}(1/\varepsilon)$ intervals for each query. Now, we leverage the fact that similar $k_j$ lie in contiguous intervals to design an efficient data structure. In particular, we can preprocess the keys in $\tilde{O}(n)$ time to ensure that we can query the sum of all values in each continuous interval of keys $\tilde{O}(1)$ time. Repeating this procedure for all queries and scaling the approximation factor (recall that our goal is to compute an $\varepsilon$-additive approximation), we obtain an algorithm that computes an $\varepsilon$-additive approximation of attention in total time $\tilde{O}(nB/\varepsilon)$. Figure 1 illustrates the rounding scheme.

The above rounding method gives a polynomial dependence on the entry bound $B$, and is only subquadratic when $B = o(n)$. Although this already improves on Alman & Song (2024a)'s algorithm (which exhibits exponential dependence on $B$, and thus only worked for values of $B = o(\sqrt{\log n})$), we would like a truly subquadratic algorithm for all polynomial $B$. To do this, we leverage the powerful polynomial method in algorithm design (see e.g. Williams (2018); Abboud et al. (2015a)).

A natural attempt to utilize the polynomial method is to approximate $e^x$ with a polynomial. As a simple case, by approximating $e^x \sim 1 + x$ we can compute $\exp(QK^T)V \sim \mathbf{1}\mathbf{1}^\top V + QK^T V$

efficiently. However, $e^x$ can only be approximated well by polynomials with degree $p$ when $|x| \leq p$ (Aggarwal & Alman, 2022). For a rank $d = O(1)$ matrix $QK^\top$, $\exp(QK^\top)$ can be approximated with a rank $2^{O(B^2)}$ matrix. Using this observation (as in Alman & Song (2024a)) one can obtain a subquadratic algorithm by assuming $B = o(\sqrt{\log n})$, but this approach falls short of obtaining sub-quadratic algorithms for polynomial $B$.

We now describe how to obtain a truly sub-quadratic algorithm by leveraging the polynomial method only on relevant indices. For simplicity, consider 1-dimensional Attention. For $x = O(t)$, there is a low-degree polynomial $P$ such that $|P(x) - \exp(x)| < \varepsilon \exp(x)$. In order to apply this approximation, we crucially use the fact that irrelevant indices are discarded, since the relevant indices have $q_i k_j$ lying within an interval of length $O(t)$. Since the probabilities are normalized, we can further assume that this interval lies around 0, allowing us to approximate $\exp$ with a polynomial. Formally, we define $c_i = \max_j q_i k_j - O(t)$ and observe that $\exp(q_i k_j)$ is proportional to $\exp(q_i k_j - c_i)$. Then, we can approximate $p_{i,j}$ which is proportional to $\exp(q_i k_j - c_i)$ with a polynomial $P$ that approximates $\exp$ on the range $O(t)$, since for all relevant indices $q_i k_j - c_i = O(t)$. We denote $\hat{p}_{i,j} \propto P(q_i k_j - c_i)$ as our approximate probabilities and output $\hat{o}_i = \sum_j \hat{p}_{i,j} v_j$. As above, if the approximate probabilities are accurate, our output is a good multiplicative approximation of attention computation.

It remains to argue that our algorithm is efficient. Note that it suffices to describe how to compute $\sum_j P(q_i k_j - c_i) v_j$ over relevant $j$ since we can compute $\hat{o}_i$ by computing this quantity twice (once with $v$ and once with $v$ replaced by $\mathbf{1}$ for normalization). The idea is that in contrast to the exponential function, the polynomial $P(q_i k_j - c_i)$ can be decoupled into a product of terms that only depend on $q_i$ and terms that only depend on $k_j$ (see Equation (2) for example). As in the rounding scheme, we use the fact that relevant keys lie in a continuous interval to create a data-structure that preprocesses the terms depending on $k_j$ in $\tilde{O}(n)$ time, while for each query $q_i$, efficiently supports queries to relevant precomputed values in $\tilde{O}(1)$ time.

**Generalizing to Higher Dimensions.** What happens when we try to generalize this algorithm to higher dimensions? In one dimension, we knew that for each $i$, the set of relevant $j$ included all $j$ where $q_i k_j \geq q_i k_{\max} - t$. In higher dimensions, our goal is similarly to compute a set of relevant indices $j$ relative to each $Q_i$ such that (1) discarding irrelevant indices outside this range does not significantly affect the additive error of our estimate and (2) the range of $Q_i \cdot K_j$ is now sufficiently restricted so that we can use a low-degree polynomial to approximate $\exp(Q_i \cdot K_j)$.

In one dimension, the set of all relevant $j$ consists exactly of the set of sufficiently large $k_j$ (either in the positive or negative direction). A simple interval searching data structure can support the necessary queries. In $d > 1$ dimensions, each row of $Q, K$ (denoted $Q_i, K_j$) is now a $d$-dimensional vector. Even in 2 dimensions, different $K_j$ may be larger with respect to different $Q_i$. Sorting all $K_j$ with respect to each $Q_i$ already requires $n^2$ time. Instead, the key observation is that the set of relevant $j$ with respect to $Q_i$ is exactly the set of $K_j$ contained in the half-space

$$\left\{ x \in \mathbb{R}^d : Q_i \cdot x \geq \max_j Q_i \cdot K_j - t \right\}.$$

This can be handled with a simplex range-searching data structure (Matoušek, 1992). In particular, we can initialize the data structure using points $\{K_j\}$ so that for each $Q_i$ we can query the data structure for the appropriate half-space. Matoušek's data structure supports queries in $\tilde{O}(n^{1-1/d})$ time and computes the sum of the weights assigned to all points in the half-space. Since in high dimensions, the number of monomials in the polynomial $P$ grows exponentially in dimension $d$, we need to instantiate and query $2^{\Omega(d)}$ instances of Matoušek's data structure. Still, for constant $d = O(1)$, this only occurs sub-polynomial factors in runtime. Using appropriate queries to the data structure over all $i$, our algorithm requires $\tilde{O}(n^{2-1/d})$ time. Figure 2 illustrates the algorithm.

**Generalizing to Low Rank Matrices.** To generalize the algorithm for low-rank matrices $Q, K$ with rank $r$, we may decompose $Q = U_Q V_Q^\top$, $K = U_K V_K^\top$ where $U_Q, V_Q, U_K, V_K$ are $n \times r$ matrices. Then, we obtain Theorem 1.2 by applying Theorem 1.1 to $Q' = U_Q$ and $K'^\top = V_Q^\top U_K V_K^\top$ which may be computed in $O(nr)$ time.

**Outline.** We give our algorithm in Section 3. The reduction from gradient computation to Attention computation is given in Appendix B. Our lower bounds are presented in Appendix C.

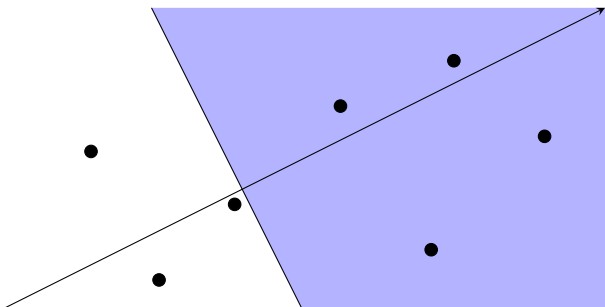

Figure 2: Algorithm for $d$-dimensional Attention illustrated for $Q_i = (2, 1)$. Relevant points are in the shaded blue region. Irrelevant points are in the white region. Weights are omitted for clarity.

## 1.2 RELATED WORK

**Approximate Attention Computation.** In an orthogonal line of work, many approximate notions of Attention have been studied to reduce its compute constraints with the goal of computing an approximation in linear time (Brown et al., 2020; Beltagy et al., 2020; Choromanski et al., 2020; Daras et al., 2020; Katharopoulos et al., 2020; Kitaev et al., 2020; Wang et al., 2020; Zaheer et al., 2020; Chen et al., 2021; Choromanski et al., 2021; Xiong et al., 2021; Gao et al., 2023a; Panigrahi et al., 2023; Malladi et al., 2023). Several works obtain provable guarantees as well as practical improvements (Zandieh et al., 2023; Han et al., 2024; Kacham et al., 2024). However, these works only obtain theoretical guarantees with respect to matrix norms such as operator norm rather than any guarantee on the correctness of each entry. Indeed, our lower bounds show that linear time approximations do not obtain such strong approximation guarantees.

In the low dimension regime $d = o(\log n)$, the Fast Multipole Method gives fast algorithms for the related Gaussian Kernel Density Estimation (KDE) problem (Alman & Guan, 2024). However, these algorithms do not apply in our regime of polynomial entries. In particular, using the standard reduction from Attention to Gaussian KDE[8] the error produced by the known KDE algorithms is amplified so that only Attention with subpolynomial entries $B = 2^{o(\log n)}$ can be computed efficiently, even with constant dimension $d = O(1)$.

**Attention with MLP Units.** Many works have studied the expressive power of Transformers (Sanford et al., 2023; 2024b;a; Yehudai et al., 2025) for classical algorithmic problems. In an independent work (Alman & Yu, 2025) show that an Attention unit with input and output MLP Layers can compute OV and (Monochromatic) Max-IP. While the constructions are similar, we reduce (Bichromatic) Max-IP to Attention, and thus obtain a strong conditional lower bound for $d = 2^{\Theta(\log^* n)}$ via (Chen, 2018).

Rather than allowing arbitrary inputs $Q, K, V \in \mathbb{R}^{n \times d}$, these works consider Attention with MLP Units: Given inputs $X \in \mathbb{R}^{n \times d_1}$ and $W_Q, W_K, W_V \in \mathbb{R}^{d_1 \times d}$, compute $Q = XW_Q, K = XW_K, V = XW_V$ and then $\text{Attention}(Q, K, V)$. This preprocessing step requires only $O(nd^2)$ time and does not change the running time of our algorithm. Via a simple modification (to either our construction or (Alman & Yu, 2025)),[9] it is possible to show that an Attention unit with MLP Units can compute (Bichromatic) Max-IP. Our reductions from OV naturally hold for bichromatic instances as well.

**Variants of Attention and Transformers** Several works have studied variants of attention and transformers (Hu et al., 2024; Ke et al., 2025), including several which leverage the polynomial method for fast computation (Alman & Song, 2023; 2025a).

**Attention Computation in Alternative Settings.** Attention has also been studied in several settings, including differential privacy (Gao et al., 2023c), fine-tuning (Hu et al., 2025), dynamic updates (Brand et al., 2023), quantum algorithms (Gao et al., 2023b), and I/O complexity (Saha & Ye, 2024).

---

[8]Map $x \mapsto (x, 0, r_x)$ and $y \mapsto (y, r_y, 0)$ for appropriate $r_x, r_y$ so that $\|x - y\|^2 = R - 2x \cdot y$ for some constant $R$.

[9]We describe how to obtain $Q, K$. Given sets of vectors $A, B \subset \mathbb{R}^d$, let $X \in \mathbb{R}^{n \times 2d}$ consist of $A$ in the first $d$ columns, $B$ in the next $d$ columns. Let $W_Q = \begin{pmatrix} I_d & 0 \end{pmatrix}$ and $W_K = \begin{pmatrix} 0 & I_d \end{pmatrix}$.

Conditional lower bounds for Attention have been studied as well (Keles et al., 2023; Alman & Song, 2024a;b; Alman & Yu, 2025).

## 2 PRELIMINARIES

We begin with the relevant definitions. Let $\log$ denote the natural log. Let $[n] = \{1, 2, \ldots, n\}$. For a matrix $M \in \mathbb{R}^{n \times m}$, we denote its $(i,j)$-entry by $M_{i,j}$, its transpose $M^\top$, and its inverse $M^{-1}$. Let $\|M\|_\infty := \max_{i,j} |M_{i,j}|$ and $\exp(M)$ denote applying $e^x$ entry-wise to $M$. Let $\mathbf{0}$ and $\mathbf{1}$ denote the all zeros and all ones vectors. For a vector $v \in \mathbb{R}^n$, $\mathrm{diag}(v)$ denotes the $n \times n$ diagonal matrix whose $(i,i)$-entry equals $v_i$.

**Fine-grained Complexity Hypotheses.** We establish new fine-grained lower bounds for the approximate attention computation problem $\mathsf{AttC}(n, d, B, \varepsilon)$. These lower bounds are conditional on some well-known fine-grained complexity hypotheses, which we introduce below.

The Strong Exponential Time Hypothesis (SETH) was introduced by Impagliazzo & Paturi (2001). They hypothesized that solving $k$-SAT for $k \geq 3$ cannot be significantly improved beyond exhaustive search.

**Hypothesis 2.1** (Strong Exponential Time Hypothesis (SETH)). *For every $\varepsilon > 0$, there is a positive integer $k \geq 3$ such that $k$-SAT on formulas with $n$ variables cannot be solved in $O(2^{(1-\varepsilon)n})$ time, even by randomized algorithms.*

SETH is a strengthening of the famous $\mathbf{P} \neq \mathbf{NP}$ conjecture and has later been used to derive fine-grained lower bounds for many fundamental computational problems, from string edit distance (Backurs & Indyk, 2018) to graph diameter (Roditty & Vassilevska Williams, 2013). Our lower bounds under SETH will proceed via reduction to the Orthogonal Vectors (OV) Problem and the Max-IP Problem.

**Theorem 2.2** (Williams (2004)). *Assuming SETH, for any $\delta > 0$ there is a constant $C$ such that any randomized algorithm solving OV in dimension $d = C \log n$ with high probability requires $\Omega(n^{2-\delta})$ time.*

The Max-IP problem asks to compute given sets of integer-valued vectors $A, B \in \mathbb{Z}^d$, $\max_{a \in A, b \in B} a \cdot b$. Chen (2018) showed that computing Max-IP requires $n^{2-o(1)}$ time even when $d = 2^{\Theta(\log^* n)}$.

**Theorem 2.3** (Chen (2018)). *Assuming SETH, for any $\delta > 0$ there is a constant $C$ such that any exact algorithm for Max-IP in dimension $d = C^{\log^* n}$ with $O(\log n)$-bit entries requires $\Omega(n^{2-\delta})$ time.*

## 3 FAST ATTENTION FOR CONSTANT HEAD DIMENSION

In this section, we present our algorithms for computing Attention in truly subquadratic time for constant head dimension $d$ and polynomial entry size $B$.

**Theorem 1.1** (Main Theorem). *Let $d = O(1)$. There is an algorithm that computes $\mathsf{AttC}(n, d, B, \varepsilon)$ in $\tilde{O}(n^{2-1/d} \cdot \mathrm{polylog}(B/\varepsilon))$ time.*

The algorithm naturally extends to the case when $d$ is large but the matrices are low dimensional. Omitted proofs in this section may be found in Appendix A.2. A key tool we require is an efficient data structure for the range searching problem.

**Definition 3.1** (Simplex Range Searching). Preprocess a weighted point set $\{(k_i, w_i)\}$ where $k_i \in \mathbb{R}^d$ and $w_i \in \mathbb{R}$ so that given any simplex query $\sigma$, the data structure returns $\sum_{k_i \in \sigma} w_i$.

Matoušek gives an efficient data structure for the simplex range searching problem. In our work, we will only query the data structure with halfspaces $\sigma$, which are special case of simplex queries (one can imagine a simplex defined by the half-space and a sufficiently large bounding box that contains all input points).

**Theorem 3.1** (Matoušek (1992)). *There is a data structure $\mathbf{RSDS}$ for the Simplex Range Searching problem for $n$ input points in $d$-dimension with $O(n \log n)$ preprocessing and $\tilde{O}(n^{1-1/d})$ query time.*

Given this data structure, we now present our algorithm for arbitrary head dimension $d$. Our inputs are $n \times d$ matrices $Q, K, V$ with entries in $[-B, B]$. Our goal is to compute the $n \times d$ output matrix $O = \text{Attention}(Q, K, V)$. We rewrite $O_{i,t} = \sum_j p_{i,j} V_{j,t}$ where $p_{i,j} = \frac{\exp(Q_i \cdot K_j)}{\sum_{j'} \exp(Q_i \cdot K_{j'})} \propto \exp(Q_i \cdot K_j)$.

**Step 1: Removing Irrelevant Keys.** We begin by showing that removing irrelevant keys does not significantly alter the quality of the approximation. Define for each $i \in [n]$ the maximum probability in the distribution $p_{i,j}$ as $p_{\max}^{(i)} = \max_j p_{i,j}$. Let $s_{\max}^{(i)}$ denote the maximum integer $s$ such that the half-space

$$\left\{ x \in \mathbb{R}^d : Q_i \cdot x \geq s \log(1 + \varepsilon) \right\}$$

contains at least one $K_j$ vector. In particular, $s_{\max}^{(i)}$ is the largest integer satisfying $\max_j Q_i \cdot K_j \geq s_{\max}^{(i)} \log(1 + \varepsilon)$. We now define relevant and irrelevant keys.

**Definition 3.2.** Let $j \in [n]$ be *irrelevant* with respect to $Q_i$ if $Q_i \cdot K_j < s_{\max}^{(i)} \log(1 + \varepsilon) - \log(n/\varepsilon)$. Otherwise $j$ is *relevant* with respect to $Q_i$. When $Q_i$ is clear, we simply say $j$ is irrelevant or relevant.

We argue that we can discard irrelevant indices.

**Lemma 3.2.** *Define* $p_{i,j}^{(r)} = \frac{p_{i,j}}{\sum_{relevant\ j} p_{i,j}}$ *if $j$ is relevant and $0$ otherwise for all $i, j \in [n]$. Let* $O_{i,t}^{(r)} = \sum_j p_{i,j}^{(r)} V_{j,t}$ *for all $i \in [n], t \in [d]$. Then* $\left| O_{i,t}^{(r)} - O_{i,t} \right| \leq 3\varepsilon B$.

**Step 2: Polynomial Approximation of Exponential.** We then show how to use polynomial approximations of $e^x$ to efficiently estimate attention. We require the following result:

**Lemma 3.3** (Aggarwal & Alman (2022); Alman & Song (2024a)). *Let $\varepsilon < 0.1$. There is a polynomial $P : \mathbb{R} \to \mathbb{R}$ of degree $g = \Theta \left( \max \left( \frac{\log(1/\varepsilon)}{\log(\log(1/\varepsilon)/B)}, B \right) \right)$ such that for all $x \in [-B, B]$, we have $|P(x) - \exp(x)| < \varepsilon \exp(x)$. Moreover, its coefficients are rationals with $\text{poly}(g)$-bit integer numerators and denominators and can be computed in $\text{poly}(g)$-time.*

Consider an entry $O_{i,t}$. We first remove irrelevant $j$ with respect to $Q_i$ and aim to approximate $O_{i,t}^{(r)}$. Recall that

$$O_{i,t}^{(r)} = \sum_j p_{i,j}^{(r)} V_{j,t} = \frac{\sum_{\text{relevant } j} \exp(Q_i \cdot K_j) V_{j,t}}{\sum_{\text{relevant } j} \exp(Q_i \cdot K_j)} = \frac{\sum_{\text{relevant } j} \exp(Q_i \cdot K_j - c(Q_i)) V_{j,t}}{\sum_{\text{relevant } j} \exp(Q_i \cdot K_j - c(Q_i))}$$

where $c(Q_i) := s_{\max}^{(i)} \log(1 + \varepsilon) - \log(n/\varepsilon)$.

By the definition of $s_{\max}^{(i)}$, we have that for all relevant $j$, $Q_i \cdot K_j - C(Q_i) \in [0, \log(n/\varepsilon) + \log(1 + \varepsilon)]$. We then invoke Lemma 3.3 to obtain a $g = \text{polylog}(n/\varepsilon)$-degree polynomial $P$ such that for all $x \in [0, \log(n/\varepsilon) + \log(1 + \varepsilon)] \subset [0, 2\log(n/\varepsilon)]$, $|P(x) - \exp(x)| \leq \varepsilon \exp(x)$. Define for relevant $j$, $\hat{p}_{i,j} \propto P(Q_i \cdot K_j - c(Q_i))$ as an approximation of $p_{i,j}^{(r)} \propto \exp(Q_i \cdot K_j - c(Q_i))$. For irrelevant $j$, set $\hat{p}_{i,j} = p_{i,j}^{(r)} = 0$. Then, define $\hat{O}_{i,t} = \sum_j \hat{p}_{i,j} V_{j,t}$. We claim $\hat{O}_{i,t}$ is a good approximation.

**Lemma 3.4.** $|\hat{O}_{i,t} - O_{i,t}| \leq 7\varepsilon B$ for all $i \in [n], t \in [d]$.

Furthermore, we present an algorithm that computes $\hat{O}$ efficiently. The key ingredient to the algorithm is the following data structure which utilizes the range searching data structure of Matoušek (1992).

**Lemma 3.5.** *Given matrices $Q, K, V \in \mathbb{R}^{n \times d}$ there exist functions $\phi_0, \ldots \phi_d$ such that any entry $\hat{O}_{i,t}$ can be computed with $g^{O(d)}$ queries to $\phi_0$ and $\phi_t$ and $g^{O(d)}$ additional time.*

*Furthermore, for each $\phi_t$ with $0 \leq t \leq d$ there is a data structure with $\tilde{O}\left( g^{O(d)} n \log n \right)$ preprocessing and $\tilde{O}\left( g^{O(d)} n^{1-1/d} \log(B/\varepsilon) \right)$ query time.*

We bound the running time of Algorithm 1.

**Lemma 3.6.** ApproxAttention *(Algorithm 1) runs in time* $\tilde{O}\left( n^{2-1/d} \cdot \text{polylog}(B/\varepsilon) \right)$.

To conclude the proof of Theorem 1.1, we run Algorithm 1 with error parameter $\varepsilon' \leq \frac{\varepsilon}{7B}$. We note that we can generalize our result to obtain an algorithm for computing Attention when the input matrices have low rank. We defer the proof to Appendix A.3.

---

**Algorithm 1** ApproxAttention$(Q, K, V)$

---

**Input**   : Matrices $Q, K, V \in [-B, B]^n$.
**Parameters**: Error parameter $\varepsilon$

**Output**  : Matrix $\hat{O}$ satisfying $\left\| \hat{O} - \mathrm{Attention}(q, k, v) \right\|_\infty \leq 7\varepsilon B$.

1 Compute $s_{\max}^{(i)}$ for all $i \in [n]$ using Theorem 3.1
2 Compute $c(Q_i) \leftarrow s_{\max}^{(i)} \log(1 + \varepsilon) - \log(n/\varepsilon)$ for all $i \in [n]$
3 Compute a $g$-degree polynomial $P(x)$ for range $[0, 2\log(n/\varepsilon)]$ using Lemma 3.3
4 Initialize the data structure for queries $\phi_t(i, \ell_1, \ldots, \ell_d)$ for all $0 \leq t \leq d$ using Lemma 3.5
5 Compute $\hat{O}_{i,t}$ for all $(i, t) \in [n] \times [d]$ using queries to Lemma 3.5
6 **return** $\hat{O}$

---

**Theorem 1.2.** *Let $r = O(1)$. There is an $\tilde{O}\left(nd + n^{2-1/r} \cdot \mathrm{polylog}(B/\varepsilon)\right)$ time algorithm computing* $\mathsf{AttC}(n, d, B, \varepsilon)$ *where $r = \min(\mathrm{rank}(Q), \mathrm{rank}(K))$.*

## 4 CONCLUSION

We conclude with some open questions. The most natural question is settling the complexity of Max-IP when $1 \ll d \ll 2^{\Theta(\log^* n)}$. We have shown several conditional lower bounds for Attention computation. Is Attention fine-grained equivalent to any well-studied problem? If such a relationship can be established, then breakthroughs on well-studied problems in fine-grained complexity can lead to breakthroughs on Attention computation. While this work focuses on characterizing the complexity of training a single Attention unit, the complexity of computing a full transformer remains open: perhaps the cost of computing many Attention units is less than computing each of them separately.

ACKNOWLEDGMENTS

The authors would like to thank Josh Alman for helpful discussions and pointing us to the Fast Multipole Method. The authors are grateful to anonymous reviewers for their valuable feedback and helpful suggestions. BS, YX, and CY were supported by NSF HDR TRIPODS Phase II grant 2217058 (EnCORE Institute). CY was supported by NSF grants 1652303, 1909046, 2112533. This work was done while SG and BH were at UCSD.

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

# A OMITTED PROOFS FOR ALGORITHMS

## A.1 WARM-UP: $d = 1$

For simplicity, we begin with our algorithm for the $d = 1$ case and explain how to generalize to the constant head dimension case later. Formally, we prove in this section the following result.

**Lemma A.1.** *There is an algorithm computing* $\mathsf{AttC}(n, 1, B, \varepsilon)$ *in* $\tilde{O}(n \cdot \mathrm{polylog}(B/\varepsilon))$ *time.*

In the above, $\mathrm{Attention}(q, k, v)$ is defined by viewing vectors $q, k, v \in \mathbb{R}^{n \times 1}$ as matrices. When $d = 1$, the input is given by vectors $q, k, v \in [-B, B]^n$. In the output vector, we hope to compute the entries

$$o_i = \frac{\sum_j e^{q_i k_j} v_j}{\sum_j e^{q_i k_j}}$$

for all $i$. Define the softmax probabilities

$$p_{i,j} = \frac{e^{q_i k_j}}{\sum_{j'} e^{q_i k_{j'}}}$$

so that $o_i = \sum_j p_{i,j} v_j$.

We begin with an overview of our algorithm. Without loss of generality, we assume $q_i \geq 0$ are non-negative for all $i$. In particular, if we compute $\mathrm{Attention}(|q|, k, v)$ and $\mathrm{Attention}(|q|, -k, v)$, where $|q|$ is a vector where we take entrywise absolute value of $q$, we can recover the entries of $\mathrm{Attention}(q, k, v)$ from the two outputs. If $q_i \geq 0$, we read the output from $\mathrm{Attention}(|q|, k, v)$ and otherwise we read the output from $\mathrm{Attention}(|q|, -k, v)$.

Let $k_{\max}$ denote the maximum value of $k$ and $p_{\max}^{(i)} = \max_j p_{i,j}$ be the corresponding maximum probability for some fixed $i$. First, we argue that we may ignore all indices where $k_j \ll k_{\max}$. Since all of these indices have exponentially small $p_{i,j}$, ignoring these indices incurs only a small additive error to the output estimate $\hat{o}_i$. Second, we argue that the remaining values of $k_j$ satisfy the property that $q_i k_j$ lie in a small range. In particular, on this range, we use the low-degree polynomial $P$ from Aggarwal & Alman (2022) to give a low-error approximation of the exponential function. Using this polynomial approximation, we instead compute

$$\hat{o}_i = \frac{\sum_j P(q_i k_j - c) v_j}{\sum_j P(q_i k_j - c)} \approx \frac{\sum_j e^{q_i k_j - c} v_j}{\sum_j e^{q_i k_j - c}} = \frac{\sum_j e^{q_i k_j} v_j}{\sum_j e^{q_i k_j}} = o_i$$

for some value $c$ that guarantees $q_i k_j - c$ lies in a bounded interval around 0 for the remaining values $k_j$.

Consider a monomial $m_\ell x^\ell$ of $P$. Then $\sum_j (q_i k_j - c)^\ell = \sum_{b=0}^{\ell} \binom{\ell}{b} (-c)^{\ell-b} q_i^b \sum_j k_j^b$. This allows to pre-compute $\sum_j k_j^b$ for all exponents $b$ in a pre-processing phase, and then efficiently compute $\hat{o}_i$ using the pre-computed values. We now describe the algorithm in more detail.

**Step 1: Removing Irrelevant Keys.** We argue that we can ignore irrelevant keys $k_j$ (Definition 3.2) with only small additive error in the estimate.

Since $q_i \geq 0$, by rearranging, note that for all irrelevant $j$, we have $q_i k_j - q_i k_{\max} \leq -\log(n/\varepsilon)$. Then, we conclude

$$\frac{p_{i,j}}{p_{\max}^{(i)}} = e^{q_i(k_j - k_{\max})} \leq \frac{\varepsilon}{n}.$$

Summing over all such indices $j$,

$$\sum_{\text{irrelevant } j} p_{i,j} \leq \sum_{\text{irrelevant } j} p_{\max}^{(i)} \frac{\varepsilon}{n} \leq \varepsilon.$$

Thus, if we define

$$p_{i,j}^{(r)} = \begin{cases} \frac{p_{i,j}}{\sum_{\text{relevant } j'} p_{i,j'}} & j \text{ is relevant,} \\ 0 & \text{o/w,} \end{cases}$$

we can obtain the guarantees for all relevant $j$

$$p_{i,j} \leq p_{i,j}^{(r)} \leq \frac{p_{i,j}}{1-\varepsilon}.$$

Then, define

$$o_i^{(r)} = \sum_j p_{i,j}^{(r)} v_j$$

so that

$$\left| o_i^{(r)} - o_i \right| \leq \left| \sum_{\text{relevant } j} (p_{i,j}^{(r)} - p_{i,j}) v_j \right| + \left| \sum_{\text{irrelevant } j} (p_{i,j}^{(r)} - p_{i,j}) v_j \right|$$

$$\leq \left| \sum_{\text{relevant } j} \frac{\varepsilon}{1-\varepsilon} p_{i,j} v_j \right| + \left| \sum_{\text{irrelevant } j} p_{i,j} v_j \right|$$

$$\leq \frac{\varepsilon}{1-\varepsilon} B + \varepsilon B$$

$$\leq 3\varepsilon B$$

where we assume $\varepsilon < \frac{1}{2}$.

**Step 2: Polynomial Approximation of Exponential.** We now show how we use polynomial approximations of $e^x$ to efficiently estimate attention.

Our goal is to approximate $o^{(r)}$:

$$o_i^{(r)} = \sum_{\text{relevant } j} p_{i,j}^{(r)} v_j = \frac{\sum_{\text{relevant } j} e^{q_i k_j} v_j}{\sum_{\text{relevant } j} e^{q_i k_j}} = \frac{\sum_{\text{relevant } j} e^{q_i k_j - c(q_i)} v_j}{\sum_{\text{relevant } j} e^{q_i k_j - c(q_i)}}$$

where $c(q_i) = q_i \cdot k_{\max} - \log(n/\varepsilon)$. In particular, we have $q_i k_j - c(q_i) \in [0, \log(n/\varepsilon)]$ for every relevant $j$.

On this interval, by Lemma 3.3, there is a polynomial $P$ of degree

$$g = O\left( \max\left( \frac{\log(1/\varepsilon)}{\log(\log(1/\varepsilon)/\log(n/\varepsilon))}, \log(n/\varepsilon) \right) \right) = O\left( \log(n/\varepsilon) \right)$$

such that $|P(x) - \exp(x)| \leq \varepsilon \exp(x)$ for all $x \in [0, \log(n/\varepsilon)]$. Then, we define $\hat{p}_{i,j} = \frac{P(q_i k_j - c(q_i))}{\sum_{\text{relevant } j'} P(q_i k_{j'} - c(q_i))}$ for relevant $j$ and $\hat{p}_{i,j} = 0$ otherwise. Next, define $\hat{o}_i = \sum_j \hat{p}_{i,j} v_j$. First, we prove the desired approximation guarantee. For all relevant $j$,

$$\frac{1-\varepsilon}{1+\varepsilon} p_{i,j}^{(r)} \leq \hat{p}_{i,j} \leq \frac{1+\varepsilon}{1-\varepsilon} p_{i,j}^{(r)}$$

so that

$$\left| \hat{o}_i - o_i^{(r)} \right| \leq B \sum_{\text{relevant } j} \left| \hat{p}_{i,j} - p_{i,j}^{(r)} \right|$$

$$\leq B \sum_{\text{relevant } j} 4\varepsilon p_{i,j}^{(r)} \leq 4\varepsilon B.$$

Combined with our previous bound using triangle inequality, we get

$$\|\hat{o} - o\|_\infty \leq \left\| \hat{o} - o^{(r)} \right\|_\infty + \left\| o^{(r)} - o \right\|_\infty \leq 7\varepsilon B. \tag{1}$$

Now, we describe how to compute $\hat{o}$ efficiently. Consider a monomial $m_\ell x^\ell$ of $P$. Then,

$$m_\ell (q_i k_j - c(q_i))^\ell = m_\ell \sum_{b=0}^{\ell} \binom{\ell}{b} q_i^b k_j^b (-c(q_i))^{\ell-b}$$

Summing over the indices $j$,

$$\sum_{\text{relevant } j} m_\ell (q_i k_j - c(q_i))^\ell = m_\ell \sum_{\text{relevant } j} \sum_{b=0}^\ell \binom{\ell}{b} q_i^b k_j^b (-c(q_i))^{\ell-b}$$

$$= m_\ell \sum_{b=0}^\ell \binom{\ell}{b} q_i^b (-c(q_i))^{\ell-b} \sum_{\text{relevant } j} k_j^b$$

Let $\phi(i,b) = \sum_{\text{relevant } j} k_j^b$ be the sum of $k_j^b$ for all $j$ relevant with respect to $q_i$. In particular,

$$\sum_{\text{relevant } j} P(q_i k_j - c(q_i)) = \sum_{\text{relevant } j} P(q_i k_j - c(q_i))$$

$$= \sum_{\text{relevant } j} \sum_\ell m_\ell (q_i k_j - c(q_i))^\ell$$

$$= \sum_\ell m_\ell \sum_{b=0}^\ell \binom{\ell}{b} q_i^b (-c(q_i))^{\ell-b} \phi(i,b).$$

Following similar computations we obtain

$$\sum_j P(q_i k_j - c(q_i)) v_j = \sum_\ell m_\ell \sum_{b=0}^\ell \binom{\ell}{b} q_i^b (-c(q_i))^{\ell-b} \phi_v(i,b) \tag{2}$$

where $\phi_v(i,b) = \sum_{\text{relevant } j} k_j^b v_j$.

The following lemmas show that we can compute $\hat{o}$ efficiently.

**Lemma A.2.** *Let $b \geq 1$ and $k_1 \geq k_2 \geq \ldots \geq k_n$. Let $q_1, \ldots, q_n$ be arbitrary. Then, $\phi(i,b), \phi_v(i,b)$ can be computed for all $i$ in time $O(n \log n)$ time.*

*Proof.* Given $b$, we can compute $\sum_{j=1}^J k_j^b$ for all $1 \leq J \leq n$ in $O(n)$ time. Then, for each $i$, we use binary search to find $J_i$, the maximum index $j$ where $k_j \geq \max_j k_j - \log(n/\varepsilon)/q_i$, i.e., $k_j$ is relevant with respect to $q_i$. Then we assign $\phi(i,b) = \sum_{j=1}^{J_i} k_j^b$. Over all $i$, this takes $O(n \log n)$ time. We can compute $\phi_v(i,b)$ similarly. $\qquad\square$

---

**Algorithm 2** VectorAttention$(q, k, v)$

---

**Input** : Vectors $q, k, v \in [-B, B]^n$.
**Parameters** : Error parameter $\varepsilon$
**Output** : Vector $\hat{o}$ satisfying $\|\hat{o} - \text{Attention}(q, k, v)\|_\infty \leq \varepsilon B$.
7  Compute a polynomial $P(x) = \sum_\ell m_\ell x^\ell$ for range $[0, \log(n/\varepsilon)]$ using Lemma 3.3.
8  Compute $k_{\max} \leftarrow \max_j k_j$ and sort $\{k_j\}$.
9  Compute $\phi(i,b), \phi_v(i,b)$ for all $1 \leq i \leq n, 1 \leq b \leq g$ using Lemma A.2.
10  **for** $1 \leq i \leq n$ **do**
11   $\quad$ Compute $\hat{o}_i \leftarrow \frac{\sum_{\text{relevant } j} P(q_i k_j - c(q_i)) v_j}{\sum_{\text{relevant } j} P(q_i k_j - c(q_i))}$ using Lemma A.3.
12  **return** $\hat{o}$

---

**Lemma A.3.** *Let $P(x) = \sum_\ell m_\ell x^\ell$ be a degree $g$-polynomial with $\text{poly}(g)$-bit coefficients. Given $q_i, \phi(i,b), \phi_v(i,b)$, there is an algorithm computing $\hat{o}_i$ in $\text{poly}(g)$ time.*

*Proof.* We recall that

$$\hat{o}_i = \sum_j \hat{p}_{i,j} v_j = \frac{\sum_j P(q_i k_j - c(q_i)) v_j}{\sum_j P(q_i k_j - c(q_i))}.$$

From Equation (2), we note

$$\sum_j P(q_i k_j - c(q_i)) v_j = \sum_\ell m_\ell \sum_{b=0}^\ell \binom{\ell}{b} q_i^b (-c(q_i))^{\ell-b} \phi_v(i,b)$$

so that given access to $\phi_v(i, b)$, we can compute the numerator in $\mathrm{poly}(g)$-time. Similarly, by accessing $\phi(i, b)$, we can compute the denominator as well. $\qquad\square$

To conclude the proof of Lemma A.1, we apply Algorithm 2 with $\varepsilon' = \frac{\varepsilon}{7B}$ so we obtain $\varepsilon$-approximation under Equation (1). In particular, the degree of the polynomial required is

$$g = O\left(\log(n/\varepsilon')\right) = O\left(\log(nB/\varepsilon)\right).$$

Then, Algorithm 2 takes time $\tilde{O}(n \cdot \mathrm{polylog}(B/\varepsilon))$.

## A.2 CONSTANT HEAD DIMENSION

We provide the omitted proofs for Theorem 1.1.

**Lemma 3.4.** $|\hat{O}_{i,t} - O_{i,t}| \le 7\varepsilon B$ for all $i \in [n], t \in [d]$.

This follows from identical arguments as to those in the one-dimensional warm-up.

**Lemma 3.5.** *Given matrices* $Q, K, V \in \mathbb{R}^{n \times d}$ *there exist functions* $\phi_0, \ldots \phi_d$ *such that any entry* $\hat{O}_{i,t}$ *can be computed with* $g^{O(d)}$ *queries to* $\phi_0$ *and* $\phi_t$ *and* $g^{O(d)}$ *additional time.*

*Furthermore, for each* $\phi_t$ *with* $0 \le t \le d$ *there is a data structure with* $\tilde{O}\left(g^{O(d)}n \log n\right)$ *preprocessing and* $\tilde{O}\left(g^{O(d)}n^{1-1/d} \log(B/\varepsilon)\right)$ *query time.*

*Proof.* Recall that $\hat{O}_{i,t} = \frac{\sum_{\text{relevant } j} P(Q_i \cdot K_j - c(Q_i)) V_{j,t}}{\sum_{\text{relevant } j} P(Q_i \cdot K_j - c(Q_i))}$ where $P$ is the polynomial of degree $g$ obtained from Lemma 3.3.

We begin with describing how to compute the numerator of $\hat{O}_{i,t}$. Suppose $P(x) = \sum_{\ell=0}^{g} m_\ell x^\ell$.

$$\sum_{\text{relevant } j} P(Q_i \cdot K_j - c(Q_i)) V_{j,t}$$

$$= \sum_{\text{relevant } j} \sum_\ell m_\ell (Q_i \cdot K_j - c(Q_i))^\ell V_{j,t}$$

$$= \sum_\ell m_\ell \sum_{\text{relevant } j} \sum_{\ell_0 + \ell_1 + \ldots + \ell_d = \ell} \binom{\ell}{\ell_0, \ell_1, \ldots, \ell_d} (-c(Q_i))^{\ell_0} \prod_{k=1}^{d} (Q_{i,k} K_{j,k})^{\ell_k} V_{j,t}$$

$$= \sum_\ell m_\ell \sum_{\ell_0 + \ell_1 + \ldots + \ell_d = \ell} \binom{\ell}{\ell_0, \ell_1, \ldots, \ell_d} (-c(Q_i))^{\ell_0} \prod_{k=1}^{d} Q_{i,k}^{\ell_k} \sum_{\text{relevant } j} \prod_{k=1}^{d} K_{j,k}^{\ell_k} V_{j,t}$$

$$= \sum_\ell m_\ell \sum_{\ell_0 + \ell_1 + \ldots + \ell_d = \ell} \binom{\ell}{\ell_0, \ell_1, \ldots, \ell_d} (-c(Q_i))^{\ell_0} \prod_{k=1}^{d} Q_{i,k}^{\ell_k} \phi_t(i, \ell_1, \ldots, \ell_d)$$

where we define the function $\phi_t(i, \ell_1, \ldots, \ell_d) = \sum_{\text{relevant } j} \prod_{k=1}^{d} K_{j,k}^{\ell_k} V_{j,t}$. Similarly, define the function

$$\phi_0(i, \ell_1, \ldots, \ell_d) = \sum_{\text{relevant } j} \prod_{k=1}^{d} K_{j,k}^{\ell_k}$$

so that

$$\sum_j P(Q_i \cdot K_j - c(Q_i)) =$$

$$\sum_\ell m_\ell \sum_{\ell_0 + \ell_1 + \ldots + \ell_d = \ell} \binom{\ell}{\ell_0, \ell_1, \ldots, \ell_d} (-c(Q_i))^{\ell_0} \prod_{k=1}^{d} Q_{i,k}^{\ell_k} \phi_0(i, \ell_1, \ldots, \ell_d).$$

The following lemma describes how to build the appropriate data structures.

**Lemma A.4.** *Let $\ell_1, \ldots, \ell_d$ be nonnegative integers. Let $0 \leq t \leq d$. Given matrices $Q, K, V$, there is a data structure with $O(nd + n \log n)$ preprocessing time that answers queries $\phi_t(i, \ell_1, \ldots, \ell_d)$ in $\tilde{O}\left(n^{1-1/d} \log(dB/\varepsilon)\right)$ time.*

*Proof.* We initialize two **RSDS** data structures using Theorem 3.1, one with unweighted point set $\{K_j\}$ and one with weighted point set $\left\{\left(K_j, \prod_{k=1}^{d} K_{j,k}^{\ell_k} V_{j,t}\right)\right\}_{j=1}^{n}$. By Theorem 3.1, this requires $O(n \log n)$ preprocessing. Computing each weight requires $O(nd)$ time.

Now, consider a query $\phi_t(i, \ell_1, \ldots, \ell_d)$ for some $i \in [n]$. We compute $s_{\max}^{(i)}$ using binary search with the first **RSDS** data structure. Since $|Q_i \cdot K_j| \leq dB^2$ there are at most $O(dB^2 / \log(1 + \varepsilon))$ values to search through. This requires $O(\log(dB/\varepsilon))$ queries which requires $\tilde{O}\left(n^{1-1/d} \log(dB/\varepsilon)\right)$ overall time by Theorem 3.1. The set of $j$ relevant to $Q_i$ is the set of $K_j$ such that $Q_i \cdot K_j \geq s_{\max}^{(i)} \log(1 + \varepsilon) - \log(n/\varepsilon)$. This can easily be captured by a simplex query with the half-space $Q_i \cdot x \geq s_{\max}^{(i)} \log(1 + \varepsilon) - \log(n/\varepsilon)$ and thus requires one query to the second **RSDS** instance. □

Our data structure for Lemma 3.5 is simply the combination of all data structures that answer queries $\phi_t(i, \ell_1, \ldots, \ell_d)$. Since $P$ is degree $g$ and $\ell_1 + \ell_2 + \ldots + \ell_d \leq \ell \leq g$, there are at most $(g + d)^{O(d)} = g^{O(d)}$ distinct tuples $\ell_1, \ldots, \ell_d$ since $d$ is a constant. In particular, we can initialize all the necessary data structures to compute queries of $\phi_t$ in $\tilde{O}\left(g^{O(d)}(nd + n \log n)\right)$ time.

We now show to compute an entry of $\hat{O}_{i,t}$. Note that numerator sums over $\ell$, tuples $\ell_0, \ldots, \ell_d$ of which there are at most $g^{O(d)}$ summands. Each summand can be computed with one query to $\phi_t$ and $g^{O(d)}$ additional time. Since the denominator can be computed similarly (instead querying $\phi_0$) the total time to compute $\hat{O}_{i,t}$ is $\tilde{O}\left(g^{O(d)}n^{1-1/d} \log(dB/\varepsilon)\right)$. □

**Lemma 3.6.** ApproxAttention *(Algorithm 1) runs in time $\tilde{O}\left(n^{2-1/d} \cdot \mathrm{polylog}(B/\varepsilon)\right)$.*

*Proof.* We now analyze the running time. From Lemma 3.3, we have

$$g = O\left(\max\left(\frac{\log(1/\varepsilon)}{\log(\log(1/\varepsilon)/\log(n/\varepsilon))}, \log(n/\varepsilon)\right)\right) = O(\log(n/\varepsilon)).$$

Then, to initialize all the necessary data structures, we invoke Lemma 3.5 a total of $d + 1$ times, thus requiring preprocessing time (recall $d$ is a constant)

$$\tilde{O}\left(n \cdot \mathrm{polylog}(1/\varepsilon)\right).$$

Then, computing all $\hat{O}_{i,t}$ requires time

$$\tilde{O}\left(n g^{O(d)}\left(n^{1-1/d} \log(B/\varepsilon)\right)\right) = \tilde{O}\left(n^{2-1/d} \cdot \mathrm{polylog}(B/\varepsilon)\right).$$

□

## A.3 GENERALIZATION TO LOW RANK MATRICES

To prove Theorem 1.2, we require the following standard result on computing a representation of low-rank matrices.

**Lemma A.5** (e.g., Hopcroft & Kannan; Roughgarden & Valiant). *Let $A$ be a $n \times d$ matrix of rank $r$ with entries in $[-B, B]$. Then, there is an $O(ndr)$ time algorithm computing an $n \times r$ matrix $U_A$ and a $d \times r$ matrix $V_A$ such that $A = U_A V_A^\top$. Furthermore, $U_A, V_A$ have entries bounded by $\mathrm{poly}(Bnd)$.*

Suppose we are given $n \times d$ input matrices $Q, K$ of rank $r_Q, r_K$ respectively. Then, we apply Lemma A.5 to compute $U_Q, V_Q, U_K, V_K$ in time $O(nd \max(r_Q, r_K)) = O(nd)$. Suppose without loss of generality $r_Q \leq r_K$. Then, we compute

$$Q' = U_Q, \quad K'^\top = V_Q^\top U_K V_K^\top$$

in time $O(r_Q r_K n) = O(n)$ and note that $Q', K'$ have entries bounded by $\mathrm{poly}(Bnd)$.

We then apply Theorem 1.1 to approximate $\text{Attention}(Q', K', V) = \text{Attention}(Q, K, V)$ which is an instance of $\text{AttC}(n, \min(r_Q, r_K), \text{poly}(Bnd), \varepsilon)$ which requires time

$$\tilde{O}\left(n^{2-1/\min(r_Q,r_K)} \cdot \text{polylog}(B/\varepsilon)\right)$$

to compute an output $\hat{O}$ such that $\left\|\hat{O} - \text{Attention}(Q, K, V)\right\|_\infty \leq \varepsilon$. This completes the proof of Theorem 1.2.

## B  THE COMPLEXITY OF ATTENTION GRADIENT COMPUTATION

In this section, we leverage our algorithm for approximate attention computation to obtain the corresponding upper bounds for approximate attention gradient computation. We begin by formalizing the notion of *attention optimization*:

**Definition B.1** (Attention Optimization). Given input matrices $A_1, A_2, A_3, E \in \mathbb{R}^{n \times d}$ and $Y \in \mathbb{R}^{d \times d}$, find a matrix $X \in \mathbb{R}^{d \times d}$ that minimizes the objective:

$$L(X) := \frac{1}{2}\left\|D(X)^{-1}AV - E\right\|_F^2,$$

where $A := \exp(A_1 X A_2^\top)$, $V := A_3 Y$, and $D(X) := \text{diag}(A\mathbf{1}_n) \in \mathbb{R}^{n \times n}$. [10]

The gradient of the objective function $L(X)$ with respect to $X$ is then used to optimize the attention mechanism by iteratively adjusting $X$ to minimize $L(X)$. Formally, we define the following approximate version of the gradient computation problem for attention optimization:

**Definition B.2** (Approximate Gradient Computation for Attention Optimization AAttLGC$(n, d, \varepsilon)$). Given $A_1, A_2, A_3, E \in [-B, B]^{n \times d}$, $Y \in [-B, B]^{d \times d}$, and $\varepsilon > 0$, compute a matrix $g \in \mathbb{R}^{d \times d}$ such that

$$\left\|g - \frac{dL(X)}{dX}\right\|_\infty \leq \varepsilon.$$

### B.1  NOTATION

Throughout this section we use the following notation. We overload the $\text{diag}$ operator. In this section, the $\text{diag}$ operator indicates turning all the non-diagonal entries to zero. The $\circ$ operator indicates entry-wise multiplication. The $\otimes$ operator denotes the Kronecker product, as defined by $Z[(i-1)n + \ell, (j-1)d + k] = X[i, j] \cdot Y[\ell, k]$ where $X, Y \in \mathbb{R}^{n \times d}$ and $Z \in \mathbb{R}^{n^2 \times d^2}$. The $\otimes_r$ operator denotes row-wise Kronecker product, as defined by $Z[i, (j-1)d + k] = X[i, j] \cdot Y[i, k]$ where $X, Y \in \mathbb{R}^{n \times d}$ and $Z \in \mathbb{R}^{n \times d^2}$. We use $e^{\langle i,j \rangle}$ as shorthand to denote $e^{a_{1_i} \cdot a_{2_j}}$, where $a_{1_i}$ and $a_{2_j}$ are rows of $A_1$ and $A_2$ respectively. If $M$ is a matrix, we use $M_i$ to denote the $i$-th row of $M$, $M_{*,i}$ to denote the $i$-th column of $M$. We use $M[i][j]$ to denote the $(i, j)$-th entry of $M$ (since our matrices have subscripts, the previous notation $M_{i,j}$ is confusing).

### B.2  UPPER BOUND ON ATTENTION BACKWARD COMPUTATION

We show that the backwards pass for approximate attention can be computed in time $\tilde{O}\left(n^{2-1/d} \cdot \text{polylog}(B/\varepsilon)\right)$ when $d = O(1)$.

**Theorem B.1** (Formal Theorem 1.3). *AAttLGC$(n, d, B, \varepsilon)$ is reducible to $O(d)$ calls to AAttC$(n, d, B, \frac{\varepsilon}{\Theta(ndB^3)})$ using $O(nd^2)$ time.*

**Corollary B.2.** *Let $d = O(1)$. There exists an algorithm that computes AAttLGC$(n, d, B, \varepsilon)$ in time $\tilde{O}\left(n^{2-1/d} \cdot \text{polylog}(B/\varepsilon)\right)$.*

*Proof of Corollary B.2.* This follows directly from Theorem B.1 and Theorem 1.1. □

---

[10] Alman & Song (2024b) scale the Attention matrix $A$ by $d$ for training efficiency, becoming $A := \exp\left(\frac{A_1 X A_2^\top}{d}\right)$. Since our algorithms scale polylogarithmically with entry size, we can safely ignore this scaling term.

*Proof of Theorem B.1.* We begin by recalling the following definitions from Alman & Song (2024b), which we will use to define the gradient computation formula.

**Definition B.3.** Let $A_1, A_2 \in \mathbb{R}^{n \times d}$ be two matrices and let $A = A_1 \otimes A_2 \in \mathbb{R}^{n^2 \times d^2}$. Let $x \in \mathbb{R}^{d^2}$ be the vectorization of the matrix $X \in \mathbb{R}^{d \times d}$ in Definition B.1. We define $A_{j_0} \in \mathbb{R}^{n \times d^2}$ to be the $n \times d^2$ size sub-block of $A$ consisting of rows $\{(j_0 - 1)n + j_1\}_{j_1=1}^n$. Let $f(x)$ be the $n \times n$ matrix whose $j_0$-th row, denoted $f(x)_{j_0}$, is given by:

$$f(x)_{j_0} := (\langle \underbrace{\exp(A_{j_0}x)}_{n \times 1}, \underbrace{1_n}_{n \times 1} \rangle^{-1} \underbrace{\exp(A_{j_0}x)}_{n \times 1})^\top.$$

Note that $f(x) = \exp(A_1 X A_2^\top) \cdot \mathrm{diag}(\exp(A_1 X A_2^\top)1_n)$. Therefore $f(x)Z$, where $Z$ is an $n \times d$ matrix, is evaluated by $\mathrm{Attention}(A_1, A_2, X)$.

**Definition B.4.** Let $Y \in \mathbb{R}^{d \times d}$ denote the matrix representation of $y \in \mathbb{R}^{d^2}$ and $Y_{*,i_0}$ indicate the $i_0$-th column of $Y$. $h(y) \in \mathbb{R}^{n \times d}$ is defined as the matrix whose $i_0$-th column is $h(y)_{i_0}$, which is defined as follows:

$$h(y)_{i_0} := \underbrace{A_3}_{n \times d} \underbrace{Y_{*,i_0}}_{d \times 1}.$$

Note that throughout this section, we occasionally use $h$ as a shorthand for $h(y)$. It is clear that $h(y)$ can be computed in $\mathsf{T}_{\mathsf{MUL}}(n, d, d)$ time.

**Definition B.5.** Let $c(x)$ be an $n \times d$ matrix defined as follows:

$$\underbrace{c(x)}_{n \times d} = \underbrace{f(x)}_{n \times n} \underbrace{h(y)}_{n \times d} - \underbrace{E}_{n \times d}.$$

We can approximate $c(y)$ by evaluating $\mathrm{Attention}(A_1 X, A_2, h(y))$ to get $f(x)h(y)$, then subtracting $E$ which takes $O(nd)$ time.

From Alman & Song (2024b) we have the following formula for attention gradient computation:

$$\frac{\mathrm{d}L(x)}{\mathrm{d}x} = A_1^\top[f(x) \circ (c(x)h(y)^\top)]A_2 - A_1^\top f(x) \, \mathrm{diag}[f(x)c(x)h(y)^\top]A_2$$

$$= A_1^\top[f(x) \circ ((f(x)h(y) - E)h(y)^\top)]A_2 - A_1^\top f(x) \, \mathrm{diag}[f(x)c(x)h(y)^\top]A_2$$

$$= A_1^\top[f(x) \circ (f(x)h(y)h(y)^\top)]A_2 - A_1^\top[f(x) \circ (Eh(y)^\top)]A_2$$

$$- A_1^\top f(x) \, \mathrm{diag}[f(x)c(x)h(y)^\top]A_2.$$

The first line comes from the characterization of the gradient as $\frac{\mathrm{d}L(x)}{\mathrm{d}x} = A_1^\top p(x)A_2$ where $p(x) = p_1(x) - p_2(x)$ (see Appendix D.4-D.6 of Alman & Song (2024b)). In the notation of Alman & Song (2024b), the first term corresponds to $p_1(x) := f(x) \circ q(x) := f(x) \circ (c(x)h(y)^\top)$. The second term corresponds to $p_2(x)$ which is an $n \times n$ matrix whose $j_0$-th column is $f(x)_{j_0} f(x)_{j_0}^\top q(x)_{j_0} := f(x)_{j_0} f(x)_{j_0}^\top c(x)h(y)_{j_0}^\top$. Note that $p_2(x) := f(x) \, \mathrm{diag}[f(x)q(x)] = f(x) \, \mathrm{diag}[f(x)c(x)h(y)^\top]$. Note that $q(x) = c(x)h(y)^\top$ is notation in Alman & Song (2024b) which we do not use here.

Let us denote

$$B_1 := [f(x) \circ (f(x)h(y)h(y)^\top)]A_2,$$

$$B_2 := [f(x) \circ (E)h(y)^\top)]A_2,$$

$$B_3 := f(x) \, \mathrm{diag}[f(x)c(x)h(y)^\top]A_2.$$

We now have the following formula which can clearly be computed in $O(nd)$ time if given $B_1, B_2$, and $B_3$:

$$\frac{\mathrm{d}L(x)}{\mathrm{d}x} = \underbrace{A_1^\top}_{d \times n} \underbrace{B_1}_{n \times d} - \underbrace{A_1^\top}_{d \times n} \underbrace{B_2}_{n \times d} - \underbrace{A_1^\top}_{d \times n} \underbrace{B_3}_{n \times d}.$$

Note that for each attention computation we perform in order to evaluate the attention gradient, we do with $\varepsilon_2 = \frac{\varepsilon}{\mathrm{poly}(d,B)n}$ additive error.

**Computing $B_3$.** Given $f(x), c(x)$, and $h(y)$, we can approximate $B_3$ using a series of matrix multiplications and attention computations, which are illustrated below in the following equations. $C_i$ denotes the intermediate matrix products from each of these matrix multiplications/attention computations. We compute an approximation of $B_3$ as follows:

$$
\begin{aligned}
B_3 &= f(x)\operatorname{diag}[\underbrace{f(x)}_{n\times n}\underbrace{c(x)}_{n\times d}h(y)^\top]A_2 \\
&= f(x)\operatorname{diag}[\underbrace{C_1}_{n\times d}\underbrace{h(y)^\top}_{d\times n}]A_2 \\
&= f(x)\underbrace{C_2}_{n\times n}\underbrace{A_2}_{n\times d} \\
&= f(x)\underbrace{C_3}_{n\times d}.
\end{aligned}
$$

We begin by computing $C_1 = f(x)c(x)$ by evaluating $\operatorname{Attention}(A_1X, A_2, c(x))$. Next, we compute $C_2 = \operatorname{diag}[C_1 h(y)^\top]$, which consists of the diagonal of the matrix product $C_1 h(y)^\top$. Since we only need the diagonal entries, this step takes $O(nd^2)$ time. We then compute $C_3 = C_2 A_2$. As $C_2$ is a diagonal matrix, this matrix multiplication can be performed in $O(nd)$ time. Finally, we compute $B_3 = f(x)C_3$ by evaluating $\operatorname{Attention}(A_1X, A_2, C_3)$.

We argue that our computed output is a good approximation of $B_3$. Let $\widetilde{B_3}$ denote the computed matrix. For any matrix $Z$, $\widetilde{Z}$ indicates an approximation of $Z$ derived by a step in our algorithm. Then,

$$
\begin{aligned}
\left\| B_3 - \widetilde{B_3} \right\|_\infty &\leq \left\| f(x)C_3 - \operatorname{AttC}(A_1X, A_2, \widetilde{C_3}) \right\|_\infty \\
&\leq \left\| f(x)C_3 - f(x)\widetilde{C_3} \right\|_\infty + \varepsilon_2 \\
&\leq \left\| C_3 - \widetilde{C_3} \right\|_\infty + \varepsilon_2 \\
&= \left\| \operatorname{diag}[C_1 h(y)^\top]A_2 - \operatorname{diag}[\widetilde{C_1} h(y)^\top]A_2 \right\|_\infty + \varepsilon_2 \\
&= \left\| \left[ \operatorname{diag}[C_1 h(y)^\top] - \operatorname{diag}[\widetilde{C_1} h(y)^\top] \right] A_2 \right\|_\infty + \varepsilon_2 \\
&\leq \|A_2\|_\infty \left\| \operatorname{diag}[C_1 h(y)^\top] - \operatorname{diag}[\widetilde{C_1} h(y)^\top] \right\|_\infty + \varepsilon_2 \\
&\leq \|A_2\|_\infty \left\| C_1 h(y)^\top - \widetilde{C_1} h(y)^\top \right\|_\infty + \varepsilon_2 \\
&\leq d\|A_2\|_\infty \|h(y)\|_\infty \left\| C_1 - \widetilde{C_1} \right\|_\infty + \varepsilon_2 \\
&\leq d\|A_2\|_\infty \|h(y)\|_\infty \left\| f(x)c(x) - \operatorname{AttC}(A_1X, A_2, \widetilde{c(x)}) \right\|_\infty + \varepsilon_2 \\
&\leq d\|A_2\|_\infty \|h(y)\|_\infty \left(\varepsilon_2 + \left\| f(x)c(x) - f(x)\widetilde{c(x)} \right\|_\infty \right) + \varepsilon_2 \\
&\leq d\|A_2\|_\infty \|h(y)\|_\infty \left(\varepsilon_2 + \left\| c(x) - \widetilde{c(x)} \right\|_\infty \right) + \varepsilon_2 \\
&\leq 2dB^2\varepsilon_2 + \varepsilon_2.
\end{aligned}
$$

Above, step 1 follows from how our algorithm approximates $B_3$, step 2 follows from our $\varepsilon_2$-error approximation of attention and the triangle inequality, step 3 follows from the fact that $f(x)$ is a stochastic matrix and distributivity of matrix multiplication, step 4 follows from our definition of $C_3$, step 5 follows from the distributivity of matrix multiplication, and step 6 follows from basic properties of the $\infty$-norm and diagonal matrices. Step 7 follows from the fact that the diag operator simply zeroes out the off-diagonal entries, making the off-diagonal elements of $C_1 h(y)^\top$ and $\widetilde{C_1} h(y)^\top$ identical. Step 8 follows from basic properties of the $\infty$-norm, step 9 follows from how our algorithm approximates $C_1$, step 10 follows from the triangle inequality and our $\varepsilon_2$ approximation of attention, step 11 follows from similar arguments as steps 9 and 10, and step 12 follows from entry bounds.

**Computing $B_1$.** We now show how to compute $B_1$. We begin by noting that $B_1 = \sum_{p=0}^{d}(f(x)(h(y)_{*,p} \otimes_r A_2)) \otimes_r (f(x)h(y))_{*,p}$, a fact we will prove later. Using this fact, we can compute $B_1$ efficiently, as illustrated in the following:

$$
\begin{aligned}
B_1 &= \sum_{p=0}^{d}(f(x)(h(y)_{*,p} \otimes_r A_2)) \otimes_r (f(x)h(y))_{*,p} \\
&= \sum_{p=0}^{d}(f(x)(\underbrace{h(y)_{*,p}}_{n\times 1} \otimes_r \underbrace{A_2}_{n\times d}) \otimes_r C_{5_{*,p}} \\
&= \sum_{p=0}^{d}(f(x)\underbrace{C_{6,p}}_{n\times d}) \otimes_r C_{5_{*,p}} \\
&= \sum_{p=0}^{d}\underbrace{C_{7,p}}_{n\times d} \otimes_r \underbrace{C_{5_{*,p}}}_{n\times 1} \\
&= \sum_{p=0}^{d}C_{8,p}.
\end{aligned}
$$

We begin by approximating $C_5 = f(x)h(y)$ by evaluating $\text{Attention}(A_1X, A_2, h(y))$. Next, for each $1 \le p \le d$, we compute the matrix $C_{6,p} = h(y)_{*,p} \otimes_r A_2$. Each matrix requires $O(nd)$ time to compute, so constructing all $d$ matrices incurs a total cost of $O(nd^2)$.

We then compute each matrix $C_{7,p} = f(x)C_{6,p}$ by evaluating $\text{Attention}(A_1X, A_2, C_{6,p})$ across all $p \in [d]$. Computing the row-wise Kronecker products $C_{8,p} = C_{7,p} \otimes_r C_{5_{*,p}}$ takes $O(nd)$ time for each $p \in [d]$, totaling $O(nd^2)$. Finally, summing over all $C_{8,p}$ requires an additional $O(nd^2)$ time.

We argue that our algorithm returns a close approximation of $B_1$. Let $\widetilde{B_1}$ indicate our computation of $B_1$. For any matrix $Z$, $\widetilde{Z}$ indicates an approximation of $Z$ derived by a step in our algorithm.

$$
\begin{aligned}
\left\|\widetilde{B_1} - B_1\right\|_{\infty} &= \left\|\sum_{p=0}^{d}\widetilde{C_{8,p}} - \sum_{p=0}^{d}C_{8,p}\right\|_{\infty} \\
&\le d\max_p\left\{\left\|\widetilde{C_{8,p}} - C_{8,p}\right\|_{\infty}\right\} \\
&\le d\max_p\left\{\left\|\widetilde{C_{7,p}} \otimes_r \widetilde{C_{5_{*,p}}} - C_{7,p} \otimes_r C_{5_{*,p}}\right\|_{\infty}\right\} \\
&\le d\max_p\left\{\left\|\widetilde{C_{7,p}} - C_{7,p}\right\|_{\infty}\left\|\widetilde{C_{5_{*,p}}} - C_{5_{*,p}}\right\|_{\infty}\right. \\
&\qquad + \left\|\widetilde{C_{7,p}} - C_{7,p}\right\|_{\infty}\left\|C_{5_{*,p}}\right\|_{\infty} \\
&\qquad + \left.\left\|\widetilde{C_{5_{*,p}}} - C_{5_{*,p}}\right\|_{\infty}\left\|C_{7,p}\right\|_{\infty}\right\} \\
&= d\max_p\left\{\left\|\text{AttC}(A_1X, A_2, C_{6,p}) - f(x)C_{6,p}\right\|_{\infty}\left\|\text{AttC}(A_1X, A_2, h(y))_{*,p} - C_{5_{*,p}}\right\|_{\infty}\right. \\
&\qquad + \left\|\text{AttC}(A_1X, A_2, C_{6,p}) - f(x)C_{6,p}\right\|_{\infty}\left\|C_{5_{*,p}}\right\|_{\infty} \\
&\qquad + \left.\left\|\text{AttC}(A_1X, A_2, h(y))_{*,p} - C_{5_{*,p}}\right\|_{\infty}\left\|C_{7,p}\right\|_{\infty}\right\} \\
&\le d\max_p\left\{\varepsilon_2^2 + \varepsilon_2\left\|(f(x)h(y))_{*,p}\right\|_{\infty} + \varepsilon_2\left\|f(x)(h(y)_{*,p} \otimes_r A_2)\right\|_{\infty}\right\} \\
&\le d\max_p\left\{\varepsilon_2^2 + \varepsilon_2\left\|h(y)\right\|_{\infty} + \varepsilon_2\left\|h(y)_{*,p} \otimes_r A_2\right\|_{\infty}\right\} \\
&\le d\max_p\left\{\varepsilon_2^2 + \varepsilon_2\left\|h(y)\right\|_{\infty} + \varepsilon_2\left\|h(y)_{*,p}\right\|_{\infty}\left\|A_2\right\|_{\infty}\right\} \\
&\le d\left(\varepsilon_2^2 + \varepsilon_2 B^2 + \varepsilon_2 B^3\right) = d\varepsilon_2^2 + d\varepsilon_2 B + d\varepsilon_2 B^2.
\end{aligned}
$$

Step 1 follows from our definition of $C_{8,p}$, step 2 follows from the triangle inequality, and step 3 follows from how we define $C_{8,p}$. Step 4 follows from analyzing the entry-wise error in the row-wise

Kronecker product. Let $a = C_{7,p}[i][j]$, $b = C_{5_{*,p}}[i][j]$, and let $e_1$ and $e_2$ denote the entry-wise approximation errors in $C_{7,p}[i][j]$ and $C_{5_{*,p}}[i][j]$, respectively. Then the approximated entry is $\widetilde{c} = (\widetilde{C_{7,p}} \otimes_r \widetilde{C_{5_{*,p}}})[i][j] = (a + e_1)(b + e_2) = ab + be_1 + ae_2 + e_1e_2$. Therefore, the entry-wise error in the approximation is $\widetilde{c} - c = be_1 + ae_2 + e_1e_2$, where $(c = C_{7,p} \otimes_r C_{5_{*,p}})[i][j]$.

Step 5 follows from how our algorithm approximates $C_{7,p}$ and $C_{5_{*,p}}$. Step 6 follows from the fact that $\widetilde{C_6} = C_6$ and our $\epsilon_2$ approximation of the attention computation. Step 7 follows from the fact that $f(x)$ is a stochastic matrix, step 8 is based on the linearity of the Kronecker product, and step 9 follows from entry bounds.

We defined $B_1 := [f(x) \circ (f(x)h(y)h(y)^\top)]A_2$. We now show We begin by noting that the format of each entry of $B_1$ is as follows, where $1 \leq i \leq n$ and $1 \leq j \leq d$:

$$B_1[i,j] = \sum_{\ell=0}^{n} \frac{e^{\langle i,\ell \rangle}}{\sum_{k=0}^{n} e^{\langle i,k \rangle}} \left[ \sum_{m=0}^{n} \frac{e^{\langle i,m \rangle}}{\sum_{k=0}^{n} e^{\langle i,k \rangle}} \sum_{p=0}^{d} h[\ell,p]h[m,p] \right] A_2[\ell,j]$$

$$= \sum_{p=0}^{d} \sum_{\ell=0}^{n} \frac{e^{\langle i,\ell \rangle}}{\sum_{k=0}^{n} e^{\langle i,k \rangle}} \left[ \sum_{m=0}^{n} \frac{e^{\langle i,m \rangle}}{\sum_{k=0}^{n} e^{\langle i,k \rangle}} h[m,p] \right] h[\ell,p] A_2[\ell,j].$$

We now compute the sum $\sum_{p=0}^{d} C_{8,p}$ and verify that

$$\left[ \sum_{p=0}^{d} C_{8,p} \right][i,j] = B_2[i,j].$$

Let $C_5 = f(x)h(y)$. For $1 \leq i \leq n$ and $1 \leq p \leq d$, we have:

$$C_5[i,p] = \sum_{m=0}^{n} \frac{e^{\langle i,m \rangle}}{\sum_{k=0}^{n} e^{\langle i,k \rangle}} h(y)[m,p].$$

Let $C_{6,p} = h(y)_{*,p} \otimes_r A_2$. For $1 \leq \ell \leq n$ and $1 \leq j \leq d$, this gives:

$$C_{6,p}[\ell,j] = h(y)[j,\ell] \cdot A_2[\ell,j].$$

We define $C_{7,p} = f(x)C_{6,p}$, so:

$$C_{7,p}[i,j] = \sum_{\ell=0}^{n} \frac{e^{\langle i,\ell \rangle}}{\sum_{k=0}^{n} e^{\langle i,k \rangle}} h(y)[j,\ell]A_2[\ell,j].$$

Let $C_{8,p} = f(x)C_{7,p} \otimes_r C_{5_{*,p}}$. Then for $1 \leq i \leq n$, $1 \leq j \leq d$:

$$C_{8,p}[i,j] = \left( \sum_{\ell=0}^{n} \frac{e^{\langle i,\ell \rangle}}{\sum_{k=0}^{n} e^{\langle i,k \rangle}} h(y)[j,\ell]A_2[\ell,j] \right) \left( \sum_{m=0}^{n} \frac{e^{\langle i,m \rangle}}{\sum_{k=0}^{n} e^{\langle i,k \rangle}} h(y)[m,p] \right)$$

$$= \sum_{\ell=0}^{n} \sum_{m=0}^{n} \frac{e^{\langle i,\ell \rangle}}{\sum_{k=0}^{n} e^{\langle i,k \rangle}} \cdot \frac{e^{\langle i,m \rangle}}{\sum_{k=0}^{n} e^{\langle i,k \rangle}} \cdot h(y)[m,p] \cdot h(y)[\ell,p] \cdot A_2[\ell,j].$$

Summing over all $p$, we recover:

$$B_2[i,j] = \sum_{p=0}^{d} C_{8,p}[i,j].$$

**Computing $B_2$.** We begin by noting that $B_2 = \sum_{p=0}^{d}[f(x)(h(y)_{*,p} \otimes_r A_2)] \otimes_r E_{*,p}$, a fact that we will prove later on. Using this fact, we use the following procedure to compute an approximation of $B_2$:

$$B_2 = \sum_{p=0}^{d} [f(x) \underbrace{(h(y)_{*,p}}_{n \times 1} \otimes_r \underbrace{A_2}_{n \times d})] \otimes_r E_{*,p}$$

$$= \sum_{p=0}^{d} [f(x) \underbrace{C_{9,p}}_{n \times d}] \otimes_r E_{*,p}$$

$$= \sum_{p=0}^{d} \underbrace{C_{10,p}}_{n \times d} \otimes_r \underbrace{E_{*,p}}_{n \times 1}$$

$$= \sum_{p=0}^{d} \underbrace{C_{11,p}}_{n \times d}.$$

We start by approximating the set of $d$ matrices, $C_{9,p} = h(y)_{*,p} \otimes_r A_2$. For each $1 \le p \le d$, computing $C_{9,p}$ takes $O(nd)$ time, so this takes $O(nd^2)$ time in total. We approximate each $C_{10,p} = f(x)C_{9,p}$ by evaluating $\text{Attention}(A_1 X, A_2, C_{9,p})$. Next, we compute all $C_{11,p} = C_{10,p} \otimes_r E_{*,p}$ which takes $O(nd^2)$ time in total. Finally, summing over $C_{11,p}$ takes $O(nd^2)$ time.

We now analyze the error from approximating $B_2$ using the method we just described. For any matrix $Z$, $\widetilde{Z}$ indicates an approximation of $Z$ derived by a step in our algorithm.

$$\left\| B_2 - \widetilde{B_2} \right\|_\infty = \left\| \sum_{p=0}^{d} C_{11,p} - \sum_{p=0}^{d} \widetilde{C_{11,p}} \right\|_\infty$$

$$\le d \max_p \left\{ \left\| C_{11,p} - \widetilde{C_{11,p}} \right\|_\infty \right\}$$

$$= d \max_p \left\{ \left\| C_{10,p} \otimes_r E_{*,p} - \widetilde{C_{10,p}} \otimes_r E_{*,p} \right\|_\infty \right\}$$

$$= d \max_p \left\{ \left\| [C_{10,p} - \widetilde{C_{10,p}}] \otimes_r E_{*,p} \right\|_\infty \right\}$$

$$\le d \max_p \left\{ \| E_{*,p} \|_\infty \left\| C_{10,p} - \widetilde{C_{10,p}} \right\|_\infty \right\}$$

$$= d \max_p \left\{ \| E_{*,p} \|_\infty \| f(x)(h(y)_{*,p} \otimes_r A_2) - \text{AttC}(A_1 X, A_2, h(y)_{*,p} \otimes_r A_2) \|_\infty \right\}$$

$$\le d \max_p \left\{ \varepsilon_2 \| E_{*,p} \|_\infty \right\}$$

$$\le d \varepsilon_2 B.$$

Above, step 1 follows from our definition of $C_{11,p}$, step 2 is follows from the triangle inequality, and step 3 follows from our definition of $C_{11,p}$. Step 4 follows from the linearity of the row-wise Kronecker product and step 5 follows from the fact that the row-wise Kronecker product scales every element in $C_{10,p}$ by an element in $E_{*,p}$. Step 6 follows from how we approximate $C_{10,p}$ in our algorithm, step 7 follows from our $\varepsilon_2$-error approximation of attention, and step 8 follows from our defined entry bounds.

We defined $B_2 := [f(x) \circ (E)h(y)^\top)] A_2$. Finally, we show that $B_2 = \sum_{p=0}^{d} [f(x)(h(y)_{*,p} \otimes_r A_2)] \otimes_r E_{*,p}$, which can be proven by showing that $B_2[i,j] = \sum_{p=0}^{d} C_{11,p}[i,j]$ for all $1 \le i \le n$ and $1 \le j \le d$. We note the following:

$$B_2[i,j] = \sum_{\ell=0}^{n} \frac{e^{\langle i, \ell \rangle}}{\sum_{k=0}^{n} e^{\langle i, k \rangle}} \left[ \sum_{p=0}^{d} E[i,p] h[\ell, p] \right] A_2[\ell, j]$$

$$= \sum_{p=0}^{d} \sum_{\ell=0}^{n} \frac{e^{\langle i, \ell \rangle}}{\sum_{k=0}^{n} e^{\langle i, k \rangle}} E[i,p] h[\ell, p] A_2[\ell, j],$$

and it is clear that the following is true:

$$C_{11,p}[i,j] = \sum_{\ell=0}^{n} \frac{e^{\langle i,\ell \rangle}}{\sum_{k=0}^{n} e^{\langle i,k \rangle}} E[i,p]h[\ell,p]A_2[\ell,j].$$

**Bounding Approximation Error.** Now all that is left is to show our procedure gives us an approximation of the gradient with $\varepsilon$ additive error. Recall that we did all the attention calculations with $\varepsilon_2 = \frac{\varepsilon}{\text{poly}(d,B)n}$ additive error. Let $\widetilde{\frac{dL(x)}{dx}}$ denote the matrix our procedure returns and let $\widetilde{c(x)}$ be the approximation of $c(x)$ given by $\text{Attention}(A_1 X, A_2, h(y))$.

$$\left\| \frac{dL(x)}{dx} - \widetilde{\frac{dL(x)}{dx}} \right\|_\infty = \left\| A_1^\top B_1 - A_1^\top B_2 - A_1^\top B_3 - (A_1^\top \sum_{p=0}^{d} C_{8,p} - A_1^\top \sum_{p=0}^{d} C_{11,p} - A_1^\top f(x)C_3) \right\|_\infty$$

$$= \left\| A_1^\top \left[ (B_1 - \sum_{p=0}^{d} C_{8,p}) + (B_2 - \sum_{p=0}^{d} C_{11,p}) + (B_3 - f(x)C_3) \right] \right\|_\infty$$

$$\leq n \left\| A_1^\top \right\|_\infty \left\| (B_1 - \sum_{p=0}^{d} C_{8,p}) + (B_2 - \sum_{p=0}^{d} C_{11,p}) + (B_3 - f(x)C_3) \right\|_\infty$$

$$\leq n \left\| A_1^\top \right\|_\infty \left[ \left\| B_1 - \sum_{p=0}^{d} C_{8,p} \right\|_\infty + \left\| B_2 - \sum_{p=0}^{d} C_{11,p} \right\|_\infty + \| B_3 - f(x)C_3 \|_\infty \right]$$

$$\leq nB\big( (d\varepsilon_2^2 + d\varepsilon_2 B + d\varepsilon_2 B^2) + d\varepsilon_2 B + (2dB^2\varepsilon_2 + \varepsilon_2) \big)$$

$$= O(ndB^3\varepsilon_2) = \varepsilon.$$

Above, steps 1 and 2 follow from definitions and rearranging terms, step 3 follows from basic properties of the $\infty$-norm, step 4 follows from the triangle inequality, and step 5 was justified previously.

$\square$

## C  NEW LOWER BOUNDS FOR ATTENTION

In this section, we prove Theorem 1.4 which shows Attention is hard even with $d = 2^{\Theta(\log^* n)}$ and Theorem 1.5 which shows that the standard algorithm is optimal for $d = \text{poly}(n)$. We begin with a generic self-reduction (Lemma C.1) that shows it suffices to prove lower bounds for Attention without normalization. We also prove Theorem C.7 which shows that Attention is hard for $d = \Omega(\log n)$ even for constant entry size.

Recall that in the attention computation $\text{Attention}(Q,K,V) = D^{-1}AV$, the diagonal matrix $D^{-1}$ applies a normalization to each row of $A$. In our reductions, however, it is necessary to work directly with the unnormalized entries of $A$. As a key lemma, we show that given oracle access to AttC with $\varepsilon$-additive error approximation, one can approximately recover the row sums of $A$ up to $O(\varepsilon)$-*multiplicative* errors, hence recovering the unnormalized entries of $A$. Specifically, if $S_i$ is the actual row sum of the $i$-th row of $A$, then the reduction computes an approximation $\hat{S}_i$ such that

$$|\hat{S}_i - S_i| < O(\varepsilon)S_i.$$

It turns out that multiplicative error approximation on the row sums is sufficient for our lower bound proofs.

**Lemma C.1.** *Let $0 < \varepsilon = O(1)$. Given matrices $Q, K \in [-B,B]^{n \times d}$ with $B \geq 1$, we can estimate the row sums of $A = \exp(QK^\top)$ up to $O(\varepsilon)$-multiplicative error in time*

$$O((\log\log n + \log(dB/\varepsilon))\mathsf{T}_{\mathsf{ATTC}}(n+1, d+1, B, \varepsilon)).$$

*Proof.* We use a parallel binary search approach to estimate the row sums. In order to implement parallel binary search, it suffices to perform the following task $\mathcal{T}$:

Given an array of numbers $\mathbf{c} = [c_1, \ldots, c_n]^\top$, output an array $\mathbf{b} \in \{0, 1\}^n$ such that if $S_i \geq (1+\varepsilon)c_i$, then $b_i = 1$; if $S_i \leq (1-\varepsilon)c_i$, then $b_i = 0$. Otherwise, $b_i$ can be arbitrary.

Indeed, at each round we let $c_i := (1+\varepsilon)^{f_i - 1}$ for some $f_i$. We use the indicator $b_i = 1$ to perform binary search for the smallest $f_i$ such that $(1+\varepsilon)^{f_i} \geq S_i$ for all $i$. Such an $f_i$ gives the guarantee that $S_i \leq (1+\varepsilon)^{f_i} < (1+\varepsilon)S_i$, which is an $\varepsilon$-multiplicative approximation of $S_i$. Note that the value of each row sum $S_i$ belongs to the range $[n \exp(-B^2 d), n \exp(B^2 d)]$, so we just need to binary search for the correct $f_i \in [\log_{1+\varepsilon}(n \exp(-B^2 d)), \log_{1+\varepsilon}(n \exp(B^2 d))]$. Therefore, the number of rounds for binary search (i.e., for performing the task $\mathcal{T}$) is given by

$$O(\log_2 \log_{1+\varepsilon}(n \exp(2B^2 d)) = O(\log \log n + \log(dB/\varepsilon)).$$

It now remains to show how to perform the task $\mathcal{T}$. We claim the following:

**Claim C.2.** *The task $\mathcal{T}$ can be completed with one oracle call to $\mathsf{AttC}(n+1, d+1, B, \varepsilon/100)$ and $O(nd)$ additional time.*

*Proof.* We create the following matrices as inputs to the oracle $\mathsf{AttC}(n+1, d+1, B, \varepsilon)$:

$$Q' := \begin{bmatrix} \ln \mathbf{c} & Q \\ 0 & \mathbf{0}_d^\top \end{bmatrix}, K' := \begin{bmatrix} 1 & \mathbf{0}_d^\top \\ \mathbf{0}_n & K \end{bmatrix}, V' := \begin{bmatrix} 0 & 0 & \cdots & 0 \\ 1 & 0 & \cdots & 0 \\ \vdots & \vdots & \ddots & \vdots \\ 1 & 0 & \cdots & 0 \end{bmatrix}.$$

Then,

$$Q'K'^\top = \begin{bmatrix} \ln \mathbf{c} & QK^\top \\ 0 & \mathbf{0}_d^\top \end{bmatrix},$$

so the $(i, 1)$-th entry of $\mathrm{Attention}(Q', K', V') = D'^{-1} A' V'$ would be

$$o_i = \frac{S_i}{c_i + S_i}.$$

Assume we have an $(\varepsilon/100)$-additive approximation of $o_i$ (denoted by $\hat{o}_i$). Then, we set $b_i = 1$ if $\hat{o}_i \geq \frac{1}{2}$ and $b_i = 0$ otherwise. We now show that all entries of $\mathbf{b}$ are correctly set. If $S_i \geq (1+\varepsilon)c_i$, then

$$\hat{o}_i \geq o_i - \varepsilon/100 \geq \frac{S_i}{c_i + S_i} - \varepsilon/100 \geq \frac{1+\varepsilon}{2+\varepsilon} - \varepsilon/100 > \frac{1}{2}.$$

On the other hand, if $S_i \leq (1-\varepsilon)c_i$, then

$$\hat{o}_i \leq o_i + \varepsilon/100 \leq \frac{S_i}{c_i + S_i} + \varepsilon/100 \leq \frac{1-\varepsilon}{2-\varepsilon} + \varepsilon/100 < \frac{1}{2}.$$

In the first inequality, we use $\frac{1+\varepsilon}{2+\varepsilon} > \frac{1}{2} + \frac{\varepsilon}{6}$ and in the second we use $\frac{1-\varepsilon}{2-\varepsilon} < \frac{1}{2} - \frac{\varepsilon}{6}$. Thus, the algorithm will output $b_i = 1$ in the former case and $b_i = 0$ in the latter case, as desired. □

This completes the proof of Lemma C.1. □

## C.1 Lower Bound for Attention with Small Head Dimension

In this section, we show via a reduction from the Max-IP problem that $\mathsf{AttC}(n, d, B, \varepsilon)$ requires $n^{2-o(1)}$ time when $d = 2^{\Omega(\log^* n)}$, $B = \mathrm{poly}(n)$, and $\varepsilon = O(1)$ additive approximation error. In particular, we note that we are able to compute Max-IP exactly even with oracle access to $\mathsf{AttC}$ that allows $\varepsilon = O(1)$ additive error.

**Lemma C.3.** *Let $\varepsilon > 0$. $\mathsf{Max\text{-}IP}(n, d, B)$ can be computed exactly in time*

$$O((\log \log n + \log(dB/\varepsilon))\mathsf{T}_{\mathsf{ATTC}}(n+1, d+1, O(B \log n), \varepsilon)).$$

*Proof.* Given a $\delta$, we choose a $C = C(\delta)$ and set $d = 2^{C \log *(n)}$. Let $\mathcal{A} = \{a_1, \ldots, a_n\}, \mathcal{B} = \{b_1, \ldots, b_n\} \subseteq \mathbb{Z}^d$ be two sets of $d$-dimensional integer-valued vectors with entries bounded by $B \geq 1$. Let $k = \ln n$ and we choose the smallest integer $C > 0$ such that

$$0.5C > 1 + \log_n(1 + \varepsilon) \quad \text{and} \quad -0.5C < \log_n(1 - \varepsilon).$$

Define the following matrices $Q, K \in \mathbb{R}^{n \times d}$:

$$Q := \begin{bmatrix} — & a_1^\top & — \\ — & a_2^\top & — \\ & \vdots & \\ — & a_n^\top & — \end{bmatrix}, \ K := kC \cdot \begin{bmatrix} — & b_1^\top & — \\ — & b_2^\top & — \\ & \vdots & \\ — & b_n^\top & — \end{bmatrix}. \tag{3}$$

By Lemma C.1, we get the $(1 \pm \varepsilon)$-multiplicative approximations of the row sums of $\exp(QK^\top)$ in time

$$O((\log \log n + \log(k^2 C^2 B^2 d / \varepsilon)) \mathsf{T}_{\mathsf{ATTC}}(n+1, d+1, kcB, \varepsilon)).$$

Here, note that $kCB = O(B \log n)$. Note that the $i$-th row sum is given by

$$S_i = \sum_{j=1}^n e^{kC(a_i \cdot b_j)} = \sum_{j=1}^n n^{C(a_i \cdot b_j)}.$$

Let $S_i'$ be the $(1 \pm \varepsilon)$-multiplicative approximation for $S_i$ and let $M_i := \max_j a_i \cdot b_j$ (note that all inner products are integers) be the maximum inner product over all vectors in $\mathcal{B}$ for a fixed $a_i \in \mathcal{A}$. We claim that $M_i$ can be recovered *exactly* by

$$M_i = \left\lfloor \frac{\log_n S_i'}{C} + 0.5 \right\rfloor.$$

Note that each non-maximum term on a single row can be bounded by $0 < n^{C(a_i \cdot b_j)} \leq n^{CM_i}$, so we can bound the row sum by

$$n^{CM_i} \leq S_i \leq n \cdot n^{CM_i} = n^{CM_i+1}.$$

Thus, applying $(1 \pm \varepsilon)$-approximation to the upper and lower bounds respectively we get

$$(1 - \varepsilon)n^{CM_i} \leq S_i' \leq (1 + \varepsilon)n^{CM_i+1}.$$

If we can show $M_i \leq (\log_n S_i')/C + 0.5 < M_i + 1$ then we are done. Indeed, using our definition for $C$ we get

$$\frac{\log_n S_i'}{C} + 0.5 \leq \frac{CM_i + 1 + \log_n(1 + \varepsilon)}{C} + 0.5 = M_i + \frac{1 + \log_n(1 + \varepsilon)}{C} + 0.5 < M_i + 1,$$

and

$$\frac{\log_n S_i'}{C} + 0.5 > \frac{CM_i + \log_n(1 - \varepsilon)}{C} + 0.5 = M_i + \frac{\log_n(1 - \varepsilon)}{C} + 0.5 > M_i.$$

$\square$

Combining the above reduction with the conditional lower bound for Max-IP (Theorem 2.3), we obtain Theorem 1.4.

**Theorem C.4** (Formal Theorem 1.4). *Fix $\varepsilon = \Theta(1)$ and $B = \mathrm{poly}(n)$. For all $\delta > 0$, there exists $C = C(\delta)$ and $d = 2^{C \log^* n}$ such that any algorithm computing $\mathsf{AttC}(n, d, B, \varepsilon)$ requires $n^{2-\delta}$ time under SETH.*

## C.2 Lower Bound for Attention with Large Head Dimension

In this section, we study the case of large head dimension where $d = \mathrm{poly}(n)$. Through a reduction from the OV problem, we show that computing $\mathsf{AAttC}(n, d, B, \varepsilon)$ requires explicitly computing the matrix product $QK^\top$ when $d = \mathrm{poly}(n)$, $B = O(\sqrt{\log n})$, and $\varepsilon = O(1)$ (additive approximation error). Furthermore, we establish a similar lower bound from the OV problem when $d = \mathrm{poly}(n)$, $B = O(1)$, and $\varepsilon = O\left(\frac{1}{\mathrm{poly}(n)}\right)$.

**Theorem C.5** (Formal Theorem 1.5). *Fix $d = \mathrm{poly}(n)$. There exists $B = O(\sqrt{\log n})$ and $\varepsilon = O(1)$ such that any algorithm computing $\mathsf{AttC}(n, d, B, \varepsilon)$ requires $\mathsf{T}_{\mathsf{MUL}}(n, d, n)^{1-o(1)}$ time under the Generalized High-Dimensional OV Hypothesis.*

We show the following lemma to prove Theorem C.5.

**Lemma C.6.** *The OV problem can be computed exactly with one call to $\mathsf{AttC}(n, d, B = O(\sqrt{\log n}), \varepsilon = O(1))$ and $O(nd)$ additional time.*

*Proof.* Let $\mathcal{A} = \{a_1, \ldots, a_n\}, \mathcal{B} = \{b_1, \ldots, b_n\} \subseteq \{0, 1\}^d$ be two sets of vectors. We chose a constant $c$ such that $\varepsilon < c < 1$ and a constant $k$ such that $k < \frac{1-c}{n(1+c)}$. We then define $Q, K \in \mathbb{R}^{n \times d}$:

$$
Q := -\sqrt{|\ln k|} \cdot \begin{bmatrix} — & a_1^\top & — \\ — & a_2^\top & — \\ & \vdots & \\ — & a_n^\top & — \end{bmatrix}, \quad K := \sqrt{|\ln k|} \cdot \begin{bmatrix} — & b_1^\top & — \\ — & b_2^\top & — \\ & \vdots & \\ — & b_n^\top & — \end{bmatrix}. \tag{4}
$$

Due to Lemma C.1, we can recover the row sums of $\exp(QK^\top)$ up to $\varepsilon$-multiplicative error in $O\left((\log\log n + \log(dB/\varepsilon))\mathsf{T}_{\mathsf{ATTC}}(n+1, d+1, B, \varepsilon)\right)$ time. Let $S_i$ be the $(1 \pm \epsilon)$-approximation of the $i$-th row sum.

$$
S_i := (1 \pm \epsilon) \sum_{j=1}^n e^{\ln(k)(a_i \cdot b_j)} = (1 \pm \epsilon) \sum_{j=1}^n k^{a_i \cdot b_j},
$$

which implies

$$
(1 - \epsilon) \sum_{j=1}^n k^{a_i \cdot b_j} \leq S_i \leq (1 + \epsilon) \sum_{j=1}^n k^{a_i \cdot b_j}.
$$

If there are no orthogonal pairs of vectors in $\mathcal{A}$ and $\mathcal{B}$, then $a_i \cdot b_j$ is a positive integer for all $1 \leq i, j \leq n$. Consequently, because $0 < k < 1$, the maximum value of $k^{a_i \cdot b_j}$ is $k$. From this it follows that if there are no pairs of orthogonal vectors, all of the sums $S_i, \ldots, S_n$ will be less than $1 - c$:

$$
S_i \leq (1 + \epsilon) \sum_{j=1}^n k^{a_i \cdot b_j} \leq (1 + \epsilon)nk < \frac{(1+\epsilon)(1-c)}{(1+c)} \leq \frac{(1+c)(1-c)}{(1+c)} = 1 - c.
$$

On the other hand, when there are one or more pairs of orthogonal vectors in $\mathcal{A}$ and $\mathcal{B}$, there will be at least one $k^{a_i \cdot b_j} = 1$ and a row sum $S_i$ will exist such that $S_i \geq 1 - c$:

$$
S_i \geq (1 - \epsilon) \sum_{j=1}^n k^{a_i \cdot b_j} > (1 - \epsilon)1 \geq 1 - c.
$$

By checking for the existence of a row sum $S_i$ that is greater than or equal to $1 - c$ we can determine whether there is a pair of orthogonal vectors in $\mathcal{A}$ and $\mathcal{B}$. $\qquad\square$

We also show that when $d = \Theta(\log n)$, Attention is hard under SETH even with constant entry size $B$.

**Theorem C.7.** *For all $\delta > 0$, there exists $C = C(\delta)$, $d = C \log n$ and $\varepsilon = n^{-C}$ such that any algorithm computing $\mathsf{AttC}(n, d, \log 2, \varepsilon)$ requires $\Omega\left(n^{2-\delta}\right)$ time under SETH.*

We show the following lemma to prove Theorem C.7.

**Lemma C.8.** *The OV problem on vectors of dimension $d$ can be computed with high probability in time*

$$
\tilde{O}\left((\log n)(d + \log n)\mathsf{T}_{\mathsf{ATTC}}\left(n+1, d+1, \log 2, \frac{1}{10n2^d}\right)\right).
$$

Given the above lemma, suppose we have an algorithm computing AttC. Given a $\delta$ define $\delta' = \delta/2$ and let $C' = C'(\delta')$ and $d = C' \log n$ as required in Theorem 2.2. Then, let $\varepsilon = \frac{1}{10n2^d} = n^{-C}$ for some large constant $C = C(\delta) \geq C'$. Any algorithm computing $\mathsf{T_{ATTC}}(n+1, d+1, \log 2, \varepsilon)$ then requires $\Omega(n^{2-\delta})$ time, proving Theorem C.7.

*Proof.* Let $\mathcal{A} = \{a_1, \ldots, a_n\}, \mathcal{B} = \{b_1, \ldots, b_n\} \subseteq \{0,1\}^d$ be two sets of vectors. Define $Q, K \in \mathbb{R}^{n \times d}$ to be the matrices whose rows are formed by the vectors in $\mathcal{A}$ and $\mathcal{B}$, respectively, i.e.,

$$
Q := \log(2) \begin{bmatrix} - & a_1^\top & - \\ - & a_2^\top & - \\ & \vdots & \\ - & a_n^\top & - \end{bmatrix}, \ K := \log(2) \begin{bmatrix} - & b_1^\top & - \\ - & b_2^\top & - \\ & \vdots & \\ - & b_n^\top & - \end{bmatrix}.
$$

Note that

$$
QK^\top = \begin{bmatrix} \log(2) \cdot a_1 \cdot b_1 & \cdots & \log(2) \cdot a_1 \cdot b_n \\ \vdots & \vdots & \ddots & \vdots \\ \log(2) \cdot a_n \cdot b_1 & \cdots & \log(2) \cdot a_n \cdot b_n \end{bmatrix}.
$$

and the $i$-th row sum of $\exp(QK^\top)$ is given by $\sum_{j=1}^n 2^{a_i \cdot b_i}$. In particular, note that all row sums are integers satisfying $n \leq S_i \leq n2^d$. From Lemma C.1, we can recover the row sums up to $\frac{1}{10n2^d}$ and therefore $\frac{1}{10}$-additive error in time

$$
O\left( (\log \log n + \log(dn2^d)) \mathsf{T_{ATTC}}\left( n+1, d+1, \log 2, \frac{1}{10n2^d} \right) \right).
$$

Given the $\frac{1}{10}$-additive approximation of $S_i$, we may recover $S_i$ by rounding since they are integers. Note that $S_i \leq n2^d$ and can therefore be represented in $O(d + \log n)$ bits.

If there are no orthogonal pairs of vectors in $\mathcal{A}$ and $\mathcal{B}$, then $a_i \cdot b_j$ is a positive integer for all $1 \leq i, j \leq n$, which means $2^{a_i \cdot b_j}$ is an even number. It follows that all of the sums $S_1, \ldots, S_n$ are also even numbers.

Conversely, when an orthogonal pair of vectors exists in $\mathcal{A}$ and $\mathcal{B}$, we would like to detect this based on the sums $S_1, \ldots, S_n$ as well. Note that when $a_i \cdot b_j = 0$ we have $2^{a_i \cdot b_j} = 1$, which may potentially make the sum into an odd number. However, when there are an even number of such orthogonal pairs, the sum remains even, and we cannot distinguish from the previous case. The workaround is to use a standard sampling method, so that with high probability, we include exactly one pair of orthogonal vectors in the sample, and therefore the corresponding sum will be odd.

Fix an index $1 \leq i \leq n$ such that $a_i \in \mathcal{A}$ is orthogonal to some vector in $\mathcal{B}$. Let $b^*$ be the last vector in $\mathcal{B}$ orthogonal to $a_i$. Without loss of generality, we may assume that the zero vector $\mathbf{0}_d \notin \mathcal{A}$, since we can check this in $O(nd)$ time and immediately accepts the input if this is the case. Given $\mathbf{0}_d \notin \mathcal{A}$, we know that vector $\mathbf{1}_d$ is not orthogonal to any vector in $\mathcal{A}$. Consider the following sampling procedure:

Construct $\mathcal{B}'$ by including each vector of $\mathcal{B}$ with probability $\frac{1}{2}$ independently and padding with $\mathbf{1}_d$ to ensure $\mathcal{B}'$ has $n$ vectors. Note that with probability exactly $\frac{1}{2}$ we have that $\mathcal{B}'$ contains an odd number of orthogonal vectors to $a_i$ (i.e. $b^*$ is included with probability $\frac{1}{2}$). In particular, sampling $\mathcal{B}'$ $O(\log n)$-times allows us to detect an odd row sum with high probability.

Thus, the overall algorithm requires involves $O(\log n)$ loops, where in each loop we check for an odd row-sum using Lemma C.1. The overall time is therefore

$$
\tilde{O}\left( (\log n)(d + \log n) \mathsf{T_{ATTC}}\left( n+1, d+1, \log 2, \frac{1}{10n2^d} \right) \right).
$$

$\square$

