# OpenReview forum: "Subquadratic Algorithms and Hardness for Attention with Any Temperature"
_ICLR.cc/2026/Conference — ICLR 2026 Poster_

### Official Review · Reviewer_NyLU · 2025-10-30

**Soundness:** 3
**Presentation:** 1
**Contribution:** 3
**Rating:** 4
**Confidence:** 3

**Summary:**

The paper gives subquadratic algorithms for attention at arbitrary temperature in constant dimension (and low-rank cases), paired with nearly tight conditional lower bounds. The theoretical picture is compelling.

**Strengths:**

1.The paper successfully constructs the first true subquadratic algorithm for attention under arbitrary temperature when the dimension is constant, using a combination of relevance pruning, polynomial approximation, and halfspace range queries.
2. The authors also give matching conditional lower bounds under SETH, making the theoretical landscape nearly complete.
3. The problem is relevant of pratical and theoretical, understanding when subquadratic attention is theoretically possible has direct implications for scaling large models.

**Weaknesses:**

1. Writing quality is poor. Although the theoretical results is sound, exposition is hard to follow: section transitions are abrupt; key intuitions are buried under algebra; notation is very condusing; This substantially reduces accessibility. For example, you write too many theorems in the intro, informal veision, formal version. This is really confusing and make reader hard to find you core contribution,
2. No experiments or even small numeric demos. Given the practical relevance of attention, the absence of any empirical validation (wall-clock, memory,  comparison to Flash/approximate/low-rank/kernelized attention under the paper’s error metric) makes it impossible to assess implementability. This make the paper not much meaningful to pratical sides.

**Questions:**

1. Could the authors provide any empirical or numerical evidence (even synthetic) to confirm that the proposed algorithm exhibits subquadratic scaling for n andwith  constant d? What are the actual constant factors involved in the halfspace data structure?
2. The presentation is difficult to follow. Can the authors improve the paper writing? This would greatly help readers.

---

> ### Author Response · Authors · 2025-11-17
>
> We thank the reviewer for their thoughtful comments and appreciate the positive feedback. We are particularly encouraged by the positive response to the theoretical contributions of our algorithm, as well as the characterization of attention complexity in all regimes. We are grateful for the careful reading and will correct all minor comments and typos. In what follows, we address the reviewers’ questions.
>
> Technical Overview Writing and Presentation
>
> Thank you for the comment. Our current write-up was written to ensure that all relevant main ideas could be checked by only reading the introduction, unfortunately this has made the introduction a bit technical. We have worked hard to improve readability of the revised version, removing most technical details from Section 1.1, and welcome any further comments and suggestions on how to improve our paper in this regard. The technical details are now contained in Section 3.
>
> We also have revised our work to ensure smoother transitions. Regarding the number of results in the introduction, we wanted to ensure that all of our results were communicated in the introduction. We have revised the introduction to highlight our main contribution: Theorem 1.1.
>
> Empirical Comparison and Practical Impact
>
> Indeed, our algorithm is unlikely to be practically efficient for practical values of d (e.g. 64) due to the relatively modest \\( O(n^{1/d}) \\) savings from the range-searching data structure and the \\( g^{O(d)} \\) overhead incurred by the polynomial approximation. Moreover, there are likely significant barriers to achieving practical algorithms for such values of d: our lower bound in Theorem 1.6 shows that any algorithm for attention immediately implies an algorithm for computing Max-IP, where the best known algorithms require \\( O(n^{2 - 1/\Theta(d)}) \\) time and any improvement would yield a significant breakthrough in one of the most well-studied algorithmic problems in the theory community. Given this barrier, any algorithm for attention is likely to be impractical relative to the standard \\( O(n^2 d) \\) implementation.
>
> Thus, we emphasize that our work is primarily a theoretical contribution. We also want to emphasize that our algorithm is the first and so far only sub-quadratic algorithm for attention that scales polylogarithmically with entry size: no subquadratic algorithms were not known to exist before our work. Nevertheless, the ideas in our algorithms might lead to more practical implementations, especially because there are already optimized libraries for searching vectors with large dot products between a given vector (e.g., Facebook AI Similarity Search (FAISS)). However, based on our lower bound, all such improvements will necessarily depend on heuristics of the data distribution or allow more errors in the output, so achieving such practical algorithms is out of the scope of our paper.
>
> Finally, we believe that the theoretical study of attention is a worthwhile pursuit. Indeed, previous works at ICLR have attempted to characterize the theoretical limits of transformer computation e.g. [1] [2].
>
> [1] How to Capture Higher-order Correlations? Generalizing Matrix Softmax Attention to Kronecker Computation (Alman, Song ICLR 2024)
>
> [2] Fundamental limitations on subquadratic alternatives to transformers (Alman, Yu ICLR 2025)

---

> ### Comment · Reviewer_4qhX · 2025-11-17
> **Disagree with Reviewer NyLU**
>
> In theoretical computer science, ''truly subquadratic'' and ''subquadratic'' have very different meaning. As far as I know, there is nothing called ''true subquadratic''.
>
> I think the writing quality is pretty good. You claimed ''notation is very condusing''. I think you mean ''confusing''. Most of the notations just directly follow from two previous NeurIPS papers [1,2].
>
> Listing 4 theorems in introduction is common for purely theoretical paper. They have both upper bound results and lower bound results, obviously will have at least 2. Also, I would not say 4 is too many.
>
> In addition, you also complain many other things in terms of the writing. I suggest you to give authors some concrete examples, for example which sections you think ''section transitions are abrupt'', which lines you think ''key intuitions are buried under algebra'', which paragraphs you think ''exposition is hard to follow''. There remains a lot of time before the deadline of rebuttal, if you can give the concrete examples of bad writing place, I believe that the authors maybe able to update a version for you before the deadline.
>
> This paper itself is already strong theoretical work, it improves two purely theoretical NeurIPS papers [1,2]. I don't see why having no experiments is so important for making the decision of this paper.
>
> [1] Josh Alman, Zhao Song. The Fine-Grained Complexity of Gradient Computation for Training Large Language Models. NeurIPS 2024
>
> [2] Josh Alman, Zhao Song. Fast Attention Requires Bounded Entries. NeurIPS 2023

---

### Official Review · Reviewer_RorL · 2025-10-30

**Soundness:** 3
**Presentation:** 3
**Contribution:** 2
**Rating:** 8
**Confidence:** 2

**Summary:**

In this work the authors study fast algorithms for approximate attention computation in the transformer architecture. In particular for matrices $Q,K,V \in \mathbb{R}^{n\times d}$, the goal of attention is to compute $Attention(Q,K,V) = D^{-1}AV$ where $A = \exp(QK^T/\sqrt{d})$ and $D = diag(A\mathbb{1})$. The problem formulation they study is that assuming all entries of $Q,K,V$ lie in $[-B,B]$, the goal is to compute an $\epsilon$ entrywise approximation to $Attention(Q,K,V)$. The classical naive algorithm for computing attention approximately runs in time $O(n^2 d)$, and a previous work of Alman and Song [AS2024] showed that for $d=\Theta(\log n)$ attention can be computed in time $n^{1+o(1)}$ whenever $B = o(\sqrt{\log N})$ and required $n^{2-o(1)}$ time whenever $B=\Omega(\sqrt{\log n})$ and $d=\Theta(\log n))$ under the SETH hypothesis.

The algorithm of [AS2024] scales exponentially in $B$, and their first result makes progress towards improving this by presenting an algorithm that for $d=O(1)$ runs in time $n^{2-1/d}\cdot polylog(B/\epsilon)$. In the case when either $Q$ or $K$ have rank $r$, they show their algorithm generalizes to give a runtime of $nd + n^{2-1/r}\cdot polylog(B/\epsilon)$. Finally they show that when $d=O(1)$ this algorithm can be used to compute the gradient of the query,key and value projection matrices that project the hidden dimension to obtain the respective embeddings in time $n^{2-1/d}\cdot polylog(B/\epsilon)$. Finally they improve the previous lower bound to show that when $d= 2^{\Omega(\log^{*}n}$ and $B=poly(n)$ attention computation requires $n^{2-o(1)}$ time, and also when $d=poly(n)$ they show that matrix multiplication time is necessary for attention computation.

**Strengths:**

The paper has many strengths in particular developing an algorithm for approximate attention with a poly logarithmic dependence on $B$ when $d=O(1)$ which is a significant improvement over the previous works, and improving the lower bound for even smaller values of $d$, i.e. $d=2^{\Omega(\log^{*}n)}$.

**Weaknesses:**

One mild weakness is that the result appears to be entirely theoretical, and the algorithm not practical. However this is not a big negative as the theoretical contribution is substantial.

**Questions:**

One question I have is even though $n^{2-1/d} = n^{2-o(1)}$ for any $d=\omega(1)$ but does the algorithmic result atleast hold in this regime ? Furthermore in the previous paper near the threshold $B=\sqrt{\log n}$ does your approach give a better lower bound than currently stated, i.e. can the results be improved in this regime because the strong lower bound holds for $d=2^{\Omega(\log^{*}n)}$ but $B=poly(n)$ ?

---

> ### Author Response · Authors · 2025-11-17
>
> We thank the reviewer for their thoughtful comments and appreciate the positive feedback. We are particularly encouraged by the positive response to the theoretical contributions of our algorithm, as well as the characterization of attention complexity in all regimes. We are grateful for the careful reading and will correct all minor comments and typos. In what follows, we address the reviewers’ questions.
>
> Algorithms for Small Superconstant d
>
> Our algorithmic results hold for all values of \\( B \\) and \\( d \\). For d larger than constant, our algorithm is still correct with the  running time being  \\( O(n^{2 - 1/d} g^{O(d)}) \\) but this is  likely worse (even asymptotically) than the standard  \\( O(n^2  d) \\) time algorithm. In the regime where \\( d = o(\log n) \\), one can actually obtain sub-quadratic algorithms for all sub-polynomial \\( B = 2^{o(\log n)} \\) using fast algorithms for Gaussian Kernel Density Estimation which implement the Fast Multipole Method (see lines 339-344 and footnote 8 in the paper for a more detailed discussion).
>
> Empirical Comparison and Practical Impact
>
> Indeed, our algorithm is unlikely to be practically efficient for practical values of d (e.g. 64) due to the relatively modest \\( O(n^{1/d}) \\) savings from the range-searching data structure and the \\( g^{O(d)} \\) overhead incurred by the polynomial approximation. Moreover, there are likely significant barriers to achieving practical algorithms for such values of d: our lower bound in Theorem 1.6 shows that any algorithm for attention immediately implies an algorithm for computing Max-IP, where the best known algorithms require \\( O(n^{2 - 1/\Theta(d)}) \\) time and any improvement would yield a significant breakthrough in one of the most well-studied algorithmic problems in the theory community. Given this barrier, any algorithm for attention is likely to be impractical relative to the standard \\( O(n^2 d) \\) implementation.
>
> Thus, we emphasize that our work is primarily a theoretical contribution. We also want to emphasize that our algorithm is the first and so far only sub-quadratic algorithm for attention that scales polylogarithmically with entry size: no subquadratic algorithms were not known to exist before our work. Nevertheless, the ideas in our algorithms might lead to more practical implementations, especially because there are already optimized libraries for searching vectors with large dot products between a given vector (e.g., Facebook AI Similarity Search (FAISS)). However, based on our lower bound, all such improvements will necessarily depend on heuristics of the data distribution or allow more errors in the output, so achieving such practical algorithms is out of the scope of our paper.
>
> Finally, we believe that the theoretical study of attention is a worthwhile pursuit. Indeed, previous works at ICLR have attempted to characterize the theoretical limits of transformer computation e.g. [1] [2].
>
> [1] How to Capture Higher-order Correlations? Generalizing Matrix Softmax Attention to Kronecker Computation (Alman, Song ICLR 2024)
>
> [2] Fundamental limitations on subquadratic alternatives to transformers (Alman, Yu ICLR 2025)

---

> ### Comment · Reviewer_4qhX · 2025-11-17
> **Reference for Fast Multipole Method**
>
> I agree with the reply from the authors. This paper [1] contains the complexity of Fast Multipole Method, in case Reviewer RorL is interested in this direction.
>
> [1] Josh Alman, Timothy Chu, Aaron Schild, Zhao Song. Algorithms and Hardness for Linear Algebra on Geometric Graphs. FOCS 2020: 541-552

---

### Official Review · Reviewer_4qhX · 2025-10-31

**Soundness:** 3
**Presentation:** 3
**Contribution:** 3
**Rating:** 8
**Confidence:** 5

**Summary:**

This paper investigates the computational complexity of the Attention mechanism without temperature restrictions on the softmax operation. While prior work showed that subquadratic attention is possible only for small entry bounds $B=o(\sqrt{\log n})$ when $d=\Theta(\log n)$, this paper provides the first truly subquadratic algorithm with polylogarithmic dependence on $B$. The main contributions include: (1) an $\mathcal{O}!\left(n^{2-1/d},\mathrm{polylog}(B)\right)$ algorithm for constant head dimension $d$ using polynomial approximations and geometric range-searching data structures, (2) extensions to low-rank matrices and gradient computation, and (3) conditional lower bounds showing the algorithm is nearly optimal, with hardness results for $d=2^{\Theta(\log^* n)}$ under SETH and optimality for $d=\mathrm{poly}(n)$ under the Generalized High-Dimensional OV Hypothesis.

**Strengths:**

1. Fundamental theoretical breakthrough. The paper resolves a key open question by providing the first subquadratic attention algorithm that scales polylogarithmically (rather than exponentially) with entry size $B$, enabling efficient computation for arbitrary temperature parameters. This represents a significant advance over Alman & Song (2024a), whose algorithms only worked for $B=o(\sqrt{\log n})$.


2. Novel technical approach with elegant insights. The combination of polynomial approximation of the exponential function with Matoušek's simplex range-searching data structure is innovative. The key observation that relevant indices (those contributing significantly to softmax probabilities) form a half-space in $\mathbb{R}^d$, enabling efficient geometric queries, demonstrates deep algorithmic insight that may have broader applications.


3. Comprehensive complexity characterization. The paper provides nearly tight upper and lower bounds across multiple parameter regimes (Table 1), including a much stronger hardness result for $d=2^{\Theta(\log^* n)}$ compared to previous $d=\Omega(\log n)$, and establishing optimality of the standard algorithm for polynomial head dimensions. The reduction from gradient computation to attention (Theorem B.3) is also a clean theoretical contribution.

**Weaknesses:**

1. Limited practical applicability for realistic parameters. The algorithm only achieves subquadratic time for constant d, while practical Transformers commonly use d = 64, 128, or larger. As stated on lines 136–138, “when $d=\omega(1)$, the above algorithms requires $n^{2-o(1)}$ time,” meaning no improvement over the standard $\mathcal{O}(n^{2}d)$ algorithm for most practical settings.

2. (minor) Limited experimental validation. Without experiments, it is unclear whether the algorithm is practical even for d = 2 or d = 3, or whether the overhead from the geometric data structures outweighs the asymptotic improvements.

3. (minor) Missing relevant works. Despite the soundness and high impact of the theoretical results in this paper, there are still some works on efficient attention computation that require proper discussion [1,2,3,4,5,6].

### References

[1] Josh Alman, Zhao Song. "How to Capture Higher-order Correlations? Generalizing Matrix Softmax Attention to Kronecker Computation". ICLR 2024.

[2] Josh Alman, Zhao Song. “Fast RoPE Attention: Combining the Polynomial Method and Fast Fourier Transform”. arXiv 2505.11892.

[3] Josh Alman, Zhao Song. “Only Large Weights (And Not Skip Connections) Can Prevent the Perils of Rank Collapse”. arXiv 2505.16284.

[4] Jerry Yao-Chieh Hu, Weimin Wu, Zhuoru Li, Sophia Pi, Zhao Song, Han Liu. “On Statistical Rates and Provably Efficient Criteria of Latent Diffusion Transformers (DiTs)”. NeurIPS 2024.

[5] Jerry Yao-Chieh Hu, Maojiang Su, En-Jui Kuo, Zhao Song, Han Liu. “Computational Limits of Low-Rank Adaptation (LoRA) Fine-Tuning for Transformer Models”. ICLR 2025.

[6] Yekun Ke, Xiaoyu Li, Yingyu Liang, Zhizhou Sha, Zhenmei Shi, Zhao Song. “On Computational Limits and Provably Efficient Criteria of Visual Autoregressive Models: A Fine-Grained Complexity Analysis”. arXiv 2501.04377.

**Questions:**

1. The algorithm achieves subquadratic time only for constant $d$, yet practical Transformers use $d=64, 128,$ or even $512$ for head dimensions. Given that your hardness results show $n^{2-o(1)}$ lower bounds for $d = 2^{\Theta(\log^* n)}$, could you comment on whether any algorithmic improvements are possible for the intermediate regime where $d$ ranges from $10$ to $100$?

2. Your algorithm provides exact attention computation up to polynomial precision with $\mathcal{O}(n^{2-1/d})$ time for constant $d$, while many existing works achieve $\mathcal{O}(n)$ time using approximate attention with weaker guarantees as mentioned in lines 317-323. When would your approach be preferable to linear-time approximations?

---

> ### Author Response · Authors · 2025-11-17
>
> We thank the reviewer for their thoughtful comments and appreciate the positive feedback. We are particularly encouraged by the positive response to the theoretical contributions of our algorithm, as well as the characterization of attention complexity in all regimes. We are grateful for the careful reading and will correct all minor comments and typos. In what follows, we address the reviewers’ questions.
>
> Empirical Comparison and Practical Impact
>
> Indeed, our algorithm is unlikely to be practically efficient for practical values of d (e.g. 64) due to the relatively modest \\( O(n^{1/d}) \\) savings from the range-searching data structure and the \\( g^{O(d)} \\) overhead incurred by the polynomial approximation. Moreover, there are likely significant barriers to achieving practical algorithms for such values of d: our lower bound in Theorem 1.6 shows that any algorithm for attention immediately implies an algorithm for computing Max-IP, where the best known algorithms require \\( O(n^{2 - 1/\Theta(d)}) \\) time and any improvement would yield a significant breakthrough in one of the most well-studied algorithmic problems in the theory community. Given this barrier, any algorithm for attention is likely to be impractical relative to the standard \\( O(n^2 d) \\) implementation.
>
> Thus, we emphasize that our work is primarily a theoretical contribution. We also want to emphasize that our algorithm is the first and so far only sub-quadratic algorithm for attention that scales polylogarithmically with entry size: no subquadratic algorithms were not known to exist before our work. Nevertheless, the ideas in our algorithms might lead to more practical implementations, especially because there are already optimized libraries for searching vectors with large dot products between a given vector (e.g., Facebook AI Similarity Search (FAISS)). However, based on our lower bound, all such improvements will necessarily depend on heuristics of the data distribution or allow more errors in the output, so achieving such practical algorithms is out of the scope of our paper.
>
> Finally, we believe that the theoretical study of attention is a worthwhile pursuit. Indeed, previous works at ICLR have attempted to characterize the theoretical limits of transformer computation e.g. [1] [2].
>
> [1] How to Capture Higher-order Correlations? Generalizing Matrix Softmax Attention to Kronecker Computation (Alman, Song ICLR 2024)
>
> [2] Fundamental limitations on subquadratic alternatives to transformers (Alman, Yu ICLR 2025)
>
> Missing Relevant Works
>
> Thank you for the pointers! We discuss the relevant works in order below.
>
> [1] studies generalizations of attention that can compute higher-order correlations (e.g. 3-wise correlations). They show that with polynomial approximation, one can compute these generalizations of attention in near-linear time when entries are small. It would be interesting to see when sub-cubic algorithms exist when entries can be large.
>
> [2] considers fast computation of RoPE attention via the polynomial method and FFT. Since this computes a different attention mechanism, it pursues a question that is somewhat orthogonal to our work, which focuses on the standard attention mechanism. Similar to [1], the polynomial method is applied to obtain fast algorithms.
>
> [3] argues that large entries are necessary to prevent model collapse. This further motivates the study of fast algorithms for attention when entries can be large (i.e. attention with any temperature).
>
> [4, 5, 6] are related works regarding transformers, though the setting and techniques are more orthogonal to the questions we consider. We are happy to cite related works and include them in the revised version.
>
> Large Constant d
>
> Our reduction of Max-IP to Attention (Theorem 1.6) is valid for all dimensions \\( d \\). In particular, our lower bound implies that any improvement to Attention computation will lead to an improvement to Max-IP computation, where the best known algorithms are of the form \\( n^{2 - 1/\Theta(d)} \\). Given that Max-IP is a well studied algorithmic problem, we believe any such algorithmic improvement would be a significant breakthrough in the theory community.
>
> Linear Time Approximations
>
> Thanks for the interesting question! While many linear time approximations of attention have been considered, it is now known that no linear-time alternative can solve crucial tasks such as document similarity (“Fundamental limitations on subquadratic alternatives to transformers”, Alman, Yu ICLR 2025). In particular, in order for a transformer to capture these similarity problems, we need to compute attention, where our algorithm is optimal in certain regimes (small head dimension \\( d \\)).

---

> > ### Comment · Reviewer_4qhX · 2025-11-17
> > **Satisfied by Rebuttals**
> >
> > Thanks for your reply. Overall, I think this paper is a very strong paper. I recommend this paper to be accepted.

---

### Official Review · Reviewer_Nr7R · 2025-10-31

**Soundness:** 3
**Presentation:** 2
**Contribution:** 2
**Rating:** 4
**Confidence:** 2

**Summary:**

This work analyzes the computational complexity of approximating attention calculations. Authors propose a subquadratic algorithm particularly handling constant head dimensions with polylogarithmic scaling in input sizes, and establish matching conditional lower bounds under SETH. This work connects polynomial approximation methods with fine-grained complexity theory to clarify the efficiency limits of fast attention computation.

**Strengths:**

1. This paper proposes a sub-quadratic algorithm for fast attention calculation with arbitrary temperatures within $\tilde{O}(n^{2-1/d})$ runtime.
2. There are complete complexity characterizations from constant to polynomial head dimensions.
3. The technical approach is elegant, combining polynomial approximation with range searching.

**Weaknesses:**

1. The presentation and paper structure are not quite clear, making it hard to identify which parts are original. The technical overview could focus more on main insights or intuitions, deferring technical details to appendices.
2. The main concern of this work is lacking empirical comparisons between the proposed algorithm and existing methods (as baselines), making it difficult to evaluate their practical performance and efficiency.
3. Following 2, although the theoretical contributions are thorough, this work does not clearly explain how the derived theoretical results can be transformed to practical applications for further improvements in Transformer training or inference.

**Questions:**

1. The contents in Table 1 are confusing. Are the listed previous results upper bounds or lower bounds? In addition, there are only two citations, despite that four previous results are listed.
2. The analysis in Line 392 raises concerns on the practical value of Theorem 1.3 when the head dimension $d$ is large, since then the exponent $2 - 1/d$ approaches quadratic complexity, reducing the "sub-quadratic" benefit of the proposed method.
3. In Line 452-454, the term $g^{\mathcal{O}(d)}$ depends exponentially on the head dimension $d$, which could significantly affect query time and warrants further discussions.

---

> ### Author Response · Authors · 2025-11-17
>
> We thank the reviewer for their thoughtful comments and appreciate the positive feedback. We are particularly encouraged by the positive response to the theoretical contributions of our algorithm, as well as the characterization of attention complexity in all regimes. We are grateful for the careful reading and will correct all minor comments and typos. In what follows, we address the reviewers’ questions.
>
> Technical Overview Writing and Presentation
>
> Thank you for the comment. Our current write-up was written to ensure that all relevant main ideas could be checked by only reading the introduction, unfortunately this has made the introduction a bit technical. We have worked hard to improve readability of the revised version, removing most technical details from Section 1.1, and welcome any further comments and suggestions on how to improve our paper in this regard. The technical details are now contained in Section 3.
>
> Empirical Comparison and Practical Impact
>
> Indeed, our algorithm is unlikely to be practically efficient for practical values of d (e.g. 64) due to the relatively modest \\( O(n^{1/d}) \\) savings from the range-searching data structure and the \\( g^{O(d)} \\) overhead incurred by the polynomial approximation. Moreover, there are likely significant barriers to achieving practical algorithms for such values of d: our lower bound in Theorem 1.6 shows that any algorithm for attention immediately implies an algorithm for computing Max-IP, where the best known algorithms require \\( O(n^{2 - 1/\Theta(d)}) \\) time and any improvement would yield a significant breakthrough in one of the most well-studied algorithmic problems in the theory community. Given this barrier, any algorithm for attention is likely to be impractical relative to the standard \\( O(n^2 d) \\) implementation.
>
> Thus, we emphasize that our work is primarily a theoretical contribution. We also want to emphasize that our algorithm is the first and so far only sub-quadratic algorithm for attention that scales polylogarithmically with entry size: no subquadratic algorithms were not known to exist before our work. Nevertheless, the ideas in our algorithms might lead to more practical implementations, especially because there are already optimized libraries for searching vectors with large dot products between a given vector (e.g., Facebook AI Similarity Search (FAISS)). However, based on our lower bound, all such improvements will necessarily depend on heuristics of the data distribution or allow more errors in the output, so achieving such practical algorithms is out of the scope of our paper.
>
> Finally, we believe that the theoretical study of attention is a worthwhile pursuit. Indeed, previous works at ICLR have attempted to characterize the theoretical limits of transformer computation e.g. [1] [2].
>
> [1] How to Capture Higher-order Correlations? Generalizing Matrix Softmax Attention to Kronecker Computation (Alman, Song ICLR 2024)
>
> [2] Fundamental limitations on subquadratic alternatives to transformers (Alman, Yu ICLR 2025)
>
> Table 1
>
> Thank you for the suggestion, we have updated Table 1 to be more legible (see updated PDF). The reason the previous result columns were not cited are due to the fact that they are standard (i.e. either the standard algorithm of computing attention directly, or the trivial lower bound implied by input and output size). We have also split up the table in terms of upper bounds and lower bounds to more clearly contrast our contributions with previous work.

---

> ### Comment · Reviewer_4qhX · 2025-11-17
> **Disagree with Weakness pointed out by Reviewer Nr7R**
>
> First of all, I think this paper is super well-written. I have no troubles for understanding any contents and proof details of the paper. In addition, I verify all the proofs. They are correct to me.
>
> Second, this paper's theoretical contribution is already strong enough, and should be accepted to conference. This paper improves two previous purely theoretical work [1,2] which published at NeurIPS before. I could be wrong, but personally, I don't feel complaining experiments is fair to such solid theoretical paper.
>
> [1] Josh Alman, Zhao Song. The Fine-Grained Complexity of Gradient Computation for Training Large Language Models. NeurIPS 2024
>
> [2] Josh Alman, Zhao Song. Fast Attention Requires Bounded Entries. NeurIPS 2023

---

### Meta-Review · Area_Chair_fQoH · 2025-12-13

**Summary:**

This is a TCS paper that proves the existence of a subquadratic algorithm for Attention with constant dimension and potentially large temperature, as well as a conditional (SETH-hardness) lower bound for moderately non-constant dimensions. This complements prior work which has focused on bounded temperature and high dimension. Reviewers appreciated the theoretical content. Concerns focused on the lack of any empirical aspects (in results nor discussion) and on the clarity of presentation. Both, in the case of this particular paper, are ultimately matters of individual view and inclination. In light of the overall discussion I recommend the paper for acceptance.

**Reviewer Concerns:**

Lack of empirical implications: all reviewers, even supportive ones, have voiced this concern. The rebuttal indeed openly conceded that the algorithmic results of the paper are unlikely to be of practical use and that the paper was meant as an entirely TCS work. Ultimately this is not a matter of addressing the concern, but a matter of whether one believes purely TCS papers should be accepted at a venue like ICLR. My personal view is that they are appropriate in select cases and this paper is one of them.

Clarity of presentation: the paper was revised during to rebuttal phase to improve presentation, but it remains hard to say whether this concern can be counted as addressed. Nonetheless clarity of presentation especially in proof-heavy papers is a thorny issue and individual opinions and tastes tend to vary greatly, so I don't think it can be counted as a major objection to the paper. My own view is that the paper makes as honest an effort in presentation as can be expected from papers of this kind.

**Reviewer Scores:**

It is particularly hard to speculate in this case how reviewer scores would have changed, given the conduct of an unusually overzealous reviewer already at the rebuttal stage. The already supportive reviewers would have likely kept their score but it is difficult to say whether consensus would have been reached with the two weak rejects. Working with what I have in unusual circumstances I recommend the paper be accepted.

---

### Decision · Program_Chairs · 2026-01-26

Accept (Poster)